# Evaluation of the CMIP6 marine subtropical stratocumulus cloud albedo and its controlling factors

Bida Jian[1], Jiming Li[1*], Guoyin Wang[2], Yuxin Zhao[1], Yarong Li[1], Jing Wang[1], Min Zhang[3], and Jianping Huang[1]

[1] Key Laboratory for Semi-Arid Climate Change of the Ministry of Education, College of Atmospheric Sciences, Lanzhou University, Lanzhou, Gansu, China
[2] Department of Atmospheric and Oceanic Sciences & Institute of Atmospheric Sciences, Fudan University, Shanghai, China
[3] Inner Mongolia Institute of Meteorological Sciences, Hohhot, Inner Mongolia, China

*Correspondence to*: Jiming Li(lijiming@lzu.edu.cn)

**Abstract.** The cloud albedo at the marine subtropical stratocumulus regions plays a key role in regulating the regional energy budget. Based on 12 years of monthly data from multiple satellite datasets, the long-term, monthly and seasonal cycle of averaged cloud albedo at five stratocumulus regions were investigated to inter-compare the atmosphere-only simulations between Phase 5 and 6 of the Coupled Model Inter-comparison Project (AMIP5 and AMIP6). Statistical results showed that

the long-term regressed cloud albedos were underestimated in most AMIP6 models compared with the satellite-driven cloud albedos, and the AMIP6 models produced a similar spread of AMIP5 over all regions. The monthly averaged values and seasonal cycle of cloud albedo of AMIP6 ensemble mean showed a better correlation with the satellite-driven observation than that of AMIP5 ensemble mean. However, the AMIP6 model still failed to reproduce the values and amplitude in some regions. By employing the Modern-Era Retrospective Analysis for Research and Applications Version 2 data, this study

estimated the relative contributions of different aerosols and meteorological factors on the long-term variation of marine stratocumulus cloud albedo under different cloud liquid water path (LWP) conditions. The multiple regression models can explain ~65 % of the changes in the cloud albedo. Under the monthly mean LWP $\leq$ 65 g m$^{-2}$, dust and black carbon dominantly contributed to the changes in the cloud albedo, while dust and sulfur dioxide aerosol contributed the most under the condition of 65 g m$^{-2}$ < LWP $\leq$ 120 g m$^{-2}$. These results suggest that the parameterization of cloud-aerosol interactions is

crucial for accurately simulating the cloud albedo in climate models.

## 1 Introduction

One of the critical parameters in regulating the distribution of solar radiation in the atmosphere and surface is cloud albedo, which is the proportion of incoming solar radiation reflected by clouds (Mueller et al., 2011; Wall et al., 2018). A change in the cloud albedo over low-level clouds can cause a significant alteration in the planetary albedo (Engström et al., 2014) and

even could offset the warming caused by doubled carbon dioxide (Latham et al., 2008). Recent studies employing the cloud-system-resolving and plume models have shown that changes in the cloud albedo are largely dependent on aerosol and

meteorological conditions (Wang et al., 2011; Stuart et al., 2013; Kravitz et al., 2014). However, there are still non-neglectable uncertainties in simulations (Bender et al., 2016).

This study specifically focused on the cloud albedo in the subtropical marine stratocumulus regions as it is particularly difficult to reproduce the cloud properties by numerical models (Eyring et al., 2016), which results in a larger uncertainty in energy budget simulations and climate predictions (Wood, 2012). The subtropical marine stratocumulus regions are mainly covered by low-level clouds that usually reflect most of the solar radiation and significantly contribute to the planetary albedo (Seethala et al., 2015). In addition, the contribution of the cloud albedo to planetary albedo over these dark oceans could be tremendous compared with those over snow/ice-covered regions with a high surface albedo (Mueller et al., 2011). However, it is a challenge to accurately estimate the cloud albedo in regions where there are different types of clouds for evaluating the cloud albedo resulted from the relationship between the planetary albedo and cloud fractions at a monthly scale (Bender et al., 2011; Bender et al., 2019).

To date, climate models have continuously advanced in main physical processes, model structures and initial conditions to improve the capability to reproduce numerous observed climate events (Van Weverberg et al., 2017; Huang et al., 2018). Many studies have paid attention to understanding the cloud albedo and its controlling factors over the subtropical marine stratocumulus regions for reducing the uncertainty in models' outputs (Latham et al., 2008; Wood, 2012; Engström et al., 2014; Bender et al., 2019). The cloud albedos obtained from regressing satellite observations at five typical subtropical marine stratocumulus regions exhibited distinct characteristics, ranging from 0.32 to 0.39, and noticeable diurnal variations (Bender et al., 2011; Engström et al., 2014), which may be induced by respective aerosols and meteorological conditions at each region. For example, the southeast Atlantic stratocumulus region (Namibian) is a typical region with massive biomass burning aerosols loading (Wilcox, 2010) while a dominant aerosol loading type in the Canarian region is dust (Waquet et al., 2013). As the value of cloud albedo is usually determined by the cloud optical thickness (COT) and the solar zenith angle (Wood, 2012), the main factors (i.e., the cloud droplet number and sizes) controlling the COT may affect changes in the cloud albedo (George and Wood, 2010; Xie et al., 2013; Bender et al., 2016). Further, the cloud droplets amount and the droplets sizes are affected by cloud condensation nuclei (CCN) and cloud liquid water content (Zhao et al., 2012), it is crucial to understand the interactions in key dynamical and microphysical processes controlling CCN with regard to improving the model capacity to simulate the cloud albedo (Bender et al., 2016; Rosenfeld et al., 2019).

Regarding the microphysical processes, the aerosol-cloud-radiation interactions over the subtropical marine stratocumulus regions have been actively examined in previous studies (Wang et al., 2011; Bender et al., 2016, 2019; Zhao et al., 2018). Among them, some studies have demonstrated the effect of aerosols on the marine stratocumulus cloud albedo (Twomey effect). In other words, an increase in aerosols can result in smaller droplet sizes and more droplets, leading to a higher cloud albedo (Twomey, 1974, 1977). However, the cloud-aerosol interactions are complex and varying with aerosol types due to their different effects on clouds. Unfortunately, the Intergovernmental Panel on Climate Change currently lacks confidence in estimating the global aerosol indirect effects (Boucher et al., 2013). Furthermore, the semi-direct effects of absorbing aerosols (e.g., black carbon) are also difficult to be quantified by numerical models (Herbert et al., 2020). Given different

model experiments from the Coupled Model Intercomparison Project phase 5 (CMIP5), Frey et al. (2017) estimated the impact of anthropogenic sulfate and non-sulfate aerosol forcing on changing the cloud albedo and concluded that absorbing aerosols play a key role in offsetting the cloud brightening at a certain degree. Regarding the dynamical processes, previous studies found that the dynamical factors (e.g., vertical velocity or instability) can influence not only the vapor supersaturation,

leading to the activation of CCN (Twomey, 1959; Lu et al., 2012; Rosenfeld et al., 2019), but also the cloud droplet number and effective radius and cloud optical depth, by the entrainment and detrainment of air above the clouds (Fuchs et al., 2018; Yang et al., 2019; Scott et al., 2020). Based on satellite observations, Chen et al. (2014) investigated the effects of aerosols on marine warm clouds, they found that the response of LWP to aerosol loading strongly depends on lower tropospheric stability and free-tropospheric moisture. Besides, the free-troposphere relative humidity is also considered as a critical factor

in regulating the cloud albedo, because it is closely related to the cloud-top entrainment/drying process that influences the cloud albedo effect (Betts and Ridgway, 1989).

However, most of these studies are based on rapid cloud adjustment to study the effects of specific meteorological factors, or aerosol-cloud interactions over the marine subtropical stratocumulus regions. Few systematic studies focus on the effects of meteorological factors and various aerosol types on the cloud albedo and changes at the monthly scale. Furthermore, it is

also crucial to evaluate the performance of current climate models to accurately project the cloud albedo responses to climate change. By the intercomparison of outputs between CMIP3 and CMIP5, Engström et al. (2014) found that the regressed regional averaged cloud albedo and intermodal spread of CMIP5 in the subtropical marine stratocumulus regions are more comparable with the satellite observations compared with those of CMIP3. Given the release of up-to-date CMIP6, as in the previous study, it is necessary to systematically evaluate the performance of CMIP5 and CMIP6 in reproducing the cloud

albedo for understanding advances in the skill of climate models to resolve long-standing problems in the marine stratocumulus regions. Based on multiple satellite datasets, this study evaluated the performance of ten CMIP5/AMIP and twenty-eight CMIP6/AMIP outputs. As an essential part of CMIP experiments, the AMIP outputs forced by observed sea surface temperatures (SST) and sea ice concentrations (Eyring et al., 2016) were used in the study. By employing the reanalysis data, this study quantitatively estimated the contributions of each factor to the marine stratocumulus cloud albedo

to identify main factors dominating the long-term variations of the marine stratocumulus cloud albedo. This study will provide useful information for comprehensively understanding the impacts of different aerosol types and meteorological factors on cloud albedo changes.

The article is organized as follows. The datasets and methods are given in section 2. The comparison of performances between CMIP5 and CMIP6 is presented in Section 3.1. The impacts of different aerosol types and meteorological factors on

the cloud albedo are described in Section 3.2. Lastly, section 4 addresses the conclusions and discussion.

## 2 Datasets and Method

This study compiled multiple satellite datasets, 10 CMIP5/AMIP outputs, 28 CMIP6/AMIP outputs and reanalysis data not only to evaluate the performance of CMIP5 and CMIP6 outputs but also to investigate the variations of the cloud albedo over the typical subtropical marine stratocumulus regions. Since spatial resolutions vary with climate models, all data were
interpolated to a 1.0° × 1.0° spatial resolution and monthly temporal resolution for fairly evaluating and intercomparing the performance. The following sections provide more details on the satellite datasets, CMIP5, CMIP6, and reanalysis data.

### 2.1 CERES and MODIS

Estimating the cloud albedo requires multiple atmospheric variables such as the top-of-atmosphere (TOA) downward, upward (all-sky) shortwave fluxes, cloud liquid water path (LWP) and cloud fractions. In this study, the TOA downward and
upward shortwave fluxes were obtained from the Clouds and the Earth's Radiant Energy System (CERES, Wielicki et al., 1996) Single Scanner Footprint (SSF) monthly Ed4A datasets. The LWP and cloud fractions were obtained from the Moderate Resolution Imaging Spectro-Radiometer (MODIS; Platnick et al., 2003) collection 6.1 level 3 monthly products during the period from 2003 to 2014, i.e., MODIS MYD08_M3 (Aqua) and MOD08_M3 (Terra) products, respectively. The spatial resolutions of these products are 1.0° × 1.0°. The CERES TOA shortwave fluxes were converted from broadband
(0.2-5.0μm) radiances by applying empirical angular distribution models to correct the instrument's incomplete spectral response (Loeb et al., 2001). Then, the real-time fluxes were aggregated to produce 24-hour mean fluxes from empirical diurnal albedo models that create meteorology conditions at the over-flight time (Loeb et al., 2018). It is worth noting that the aforementioned data processing may introduce some potential uncertainties, e.g., diurnal correction error, radiance-to-flux conversion error (one standard deviation, 1σ) and instrument calibration error (1σ). For example, the uncertainty in the
monthly combined regional all-sky shortwave flux was 6.2 W m$^{-2}$(CERES_SSF1deg-Hour/Day/Month_Ed4A Data Quality Summary. 2021), where the calculation of the diurnal correction uncertainty was driven by comparisons with Geostationary Earth Radiation Budget data (Doelling et al., 2013). In addition, the cloud fraction, a fraction of MODIS cloudy pixels to the total pixels at each grid box, is determined based on daytime scenes entirely and represents all cloud phases (Platnick et al., 2003). As the CERES and MODIS instruments are both carried onboard Aqua (cross the equator local time: 1130) and Terra
(pass the equator local time: 1030) satellites in polar orbits, we averaged the Aqua and Terra products to obtain the observed combined all-sky albedo, cloud fraction, LWP and cloud albedo as in the works of the Engström et al. (2015) and Bender et al. (2017). Time representation errors can be caused by the averaged cloud observations at two time points to represent the daily average. However, recent studies found that the time representation error was significant for short-term studies (up to 14%) while negligible for long-term statistical analysis (Wang and Zhao, 2017; Zhao et al. 2019a).

### 2.2 CMIP5/AMIP and CMIP6/AMIP

The outputs of 10 CMIP5/AMIP and 28 CMIP6/AMIP include all variables necessary to estimate the cloud albedo, i.e., monthly mean TOA downward, upward (all-sky) fluxes and total cloud fractions (Taylor et al., 2012; Eyring et al., 2016). This study used ten climate models that provide both CMIP5 and CMIP6 outputs and implemented the intercomparison of performance for the regressed cloud albedo during the period from 2003 to 2008. Furthermore, this study evaluated the cloud

albedo for twenty-eight CMIP6/AMIP outputs during the period from 2003 to 2014. Tables 1-2 show the characteristics of CMIP5 and CMIP6 models. Note that there is a considerable discrepancy in the total cloud fractions between the CMIP models and MODIS observations, which is caused by different definitions, cloud overlap algorithms and different threshold assumptions for cloud formation (Engström et al., 2015). Moreover, the total cloud fractions in the climate models are usually calculated based on daytime and nighttime cloud fractions, while the observed cloud fractions are only for the

daytime. As used in Engström et al. (2015), this study also employed the total cloud fractions as there are no available MODIS simulator outputs for CMIP6. Although uncertainty in cloud fraction remains, a previous study also demonstrated that the time representation error was negligible for long-term statistical analysis (Wang and Zhao, 2017).

### 2.3 MERRA-2

The study employed the Modern-Era Retrospective Analysis for Research and Applications Version 2 (MERRA-2) which

provides a long-term aerosol and atmospheric reanalysis record (1980-present) at $0.625° \times 0.5°$ resolution based on the Goddard Earth Observing System Model, version 5.12.4 (Gelaro et al., 2017). The aerosol reanalysis has been produced by a global data assimilation system that combines satellite- and ground-based observed aerosols with meteorological conditions. Here, the mass mixing ratios of different aerosol types and air density at different levels from the 3-hourly aerosol product (inst3_3d_aer_Nv) and meteorological data from monthly atmosphere product (instM_3d_asm_Np and instM_2d_asm_Nx)

were collected to represent the monthly regional aerosol and meteorological conditions. The outputs of MERRA-2 reanalysis were used during the consistent period from 2003 to 2014 with satellite observations record. As selected in McCoy et al. (2017) and Li et al. (2018), the impacts of different aerosol types on marine stratocumulus cloud albedo were evaluated based on the mass concentrations of hydrophilic black carbon (BC), hydrophilic organic carbon (OC), sulfate aerosol (SO4), sulfur dioxide (SO2), the smallest particles dust (DU, i.e., 0.1-1 μm size) and sea salt (SS, 0.03-0.1 μm size) at the 910hPa

level. The meteorological variables include the monthly vertical velocity at 900hPa and 700hPa (omega900 and omega700), surface pressure, relative humidity at 700hPa (RH700), air temperature, the eastward wind, the northward wind and surface skin temperature data. In addition, the estimated inversion strength (EIS) and horizontal temperature advection at the surface (SSTadv) factors were also calculated. Finally, all of these meteorological parameters were used to investigate the meteorological effects on the cloud albedo. The units for aerosol mass concentrations, relative humidity, vertical velocity,

EIS and SSTadv are kg m$^{-3}$, %, Pa s$^{-1}$, K and K s$^{-1}$, respectively.

### 2.4 Methods

The planetary albedo ($\alpha$) can be calculated mainly from the cloud fraction $f$ (Bender et al., 2011) as expressed in Eq. (1):

$$\alpha = \alpha_{cloud} f + \alpha_{clear}(1-f) \tag{1}$$

where, $\alpha_{cloud}$ and $\alpha_{clear}$ denote the albedo under cloudy-sky and clear-sky conditions, respectively. For a given region where the cloud and surface type are homogeneous (i.e., constant $\alpha_{cloud}$ and $\alpha_{clear}$), namely, a change in $\alpha$ should be driven by a change in the cloud fraction $f$. The cloud albedo can be estimated by the derivative of Eq. (1) as in Eq. (2):

$$\alpha_{cloud} = d\alpha / df + \alpha_{clear} \tag{2}$$

The invariable $\alpha_{cloud}$ and $\alpha_{clear}$ should be applied for the same cloud type and ocean region. In this light, as the works in Klein and Hartmann (1993), this study also analyzed only five marine stratocumulus regions: Peruvian (10°S–20°S, 80°W–90°W; A1), Namibian (10°S-20°S, 0°E-10°E; A2), Californian (20°N-30°N, 120°W-130°W; A3), Australian (25°S-35°S, 95°E-105°E; A4) and Canarian (15°N-25°N, 25°W-35°W; A5). Previous study (Engström et al., 2014) has also demonstrated that there is a near-linear relationship between cloud cover and planetary albedo in these regions. Fig. 1 illustrates the locations of the above stratocumulus regions and the near-global distribution of combined planetary albedo averaged from Aqua and Terra during the common period from 2003 to 2014. Here, EIS is defined in Wood and Bretherton (2006):

$$EIS = LTS - \Gamma_m^{850}(Z_{700} - Z_{LCL}) \tag{3}$$

where the lower-tropospheric stability (LTS) is defined as the difference in potential temperature between 700 hPa and the surface, $\Gamma_m^{850}$ is the moist-adiabatic lapse rate at 850 hPa, $Z_{700}$ and $Z_{LCL}$ is the height of the 700 hPa level and the lifting condensation level relative to the surface, respectively. As in Wood and Bretherton (2006), we assumed the surface relative humidity of 80 % to simplify the calculation of surface dew point temperature. $Z_{LCL}$ was caculated based on the method of Georgakakos and Bras (1984). In addition, SSTadv was obtained by Eq. (4) as in Qu et al., (2015):

$$SSTadv = -\frac{u}{R_E\cos\phi}\frac{\partial SST}{\partial\lambda} - \frac{v}{R_E}\frac{\partial SST}{\partial\phi} \tag{4}$$

where u and v represent the eastward and northward horizontal wind components at 1000hPa, respectively. $\phi$ and $\lambda$ are latitude and longitude, respectively. $R_E$ is the mean Earth radius and SST is the surface skin temperature. A positive/negative SSTadv indicates warm/cold advection. The SSTadv can affect the moisture transport within the cloud layer by influencing the surface sensible and latent heat fluxes, and consequently, influence the thickness of marine stratocumulus clouds (George and Wood, 2010).

In the study, to avoid influence from a seasonal cycle, the long-term mean analyses are implemented with deseasonalized monthly mean data processed by removing a mean seasonal cycle, and then adding the monthly mean value to the interannual anomalies data. The selection of variables is a crucial step to build a multiple linear regression model of the monthly cloud albedo as a function of meteorological factors and aerosol types under two different LWP scenarios (LWP < 65 g m$^{-2}$ and 65 g m$^{-2}$ < LWP < 120 g m$^{-2}$). This study selected suitable variables based on correlation analysis. If the correlation between the cloud albedo and a candidate is significant at a 90% confidence level, the variable was considered as a predictor factor. Furthermore, the partial least squares were used to reduce the collinearity between the selected variables (McCoy et al., 2017). The regression model of cloud albedo $\alpha_{cloud}$ is as follows:

$\quad \alpha_{\text{cloud}} = \sum_{i=1}^{I} a_i M_i + \sum_{j=1}^{J} b_j \log_{10} A_j + c$ (5)

where a and b are regression coefficients, c is a constant term, $M_i$ represents the ith meteorological predictor, $I$ is the number of meteorological predictor variables, $A_j$ is the jth aerosols predictor, and $J$ is the number of aerosol predictor variables.

The relative contributions of each predictor to the change in the cloud albedo (Huang and Yi, 1991) were evaluated using Eq. (6):

$\quad R_j = \frac{1}{m} \sum_{i=1}^{m} [T_{ij}^2 / (\sum_{j=1}^{a} T_{ij}^2)]$ (6)

where $m$ is the number of the monthly samples, $a$ is the number of predictors, and $T_{ij}$ is the product of the regression coefficients of each term ($b_j$) and predictor variables ($x_{ij}$).

After removing the effect of meteorological factors, we further investigated the pure relationship between aerosols and the cloud albedo using the partial correlations between $\alpha_{cloud}$ and $\log_{10}A$, as expressed in Eq. (7):

$\quad r_{\alpha_{\text{cloud}} \log_{10} A \cdot M} = \frac{r_{\alpha_{\text{cloud}} \cdot \log_{10}A} - r_{\alpha_{\text{cloud}} \cdot M} r_{\log_{10}A \cdot M}}{\sqrt{1 - r_{\alpha_{\text{cloud}} \cdot M}^2} \sqrt{1 - r_{\log_{10}A \cdot M}^2}}$ (7)

where $r_{\alpha_{\text{cloud}} \cdot \log_{10}A}$, $r_{\alpha_{\text{cloud}} \cdot M}$ and $r_{\log_{10}A \cdot M}$ is the total correlation between each variable pair and $r_{\alpha_{\text{cloud}} \log_{10}A \cdot M}$ is the correlation between $\alpha_{\text{cloud}}$ and $\log_{10}A$ which eliminates the effects of meteorological factors M. More details on the partial correlation are described in Jiang et al. (2018) and Engström and Ekman (2010).

**3 Results**

**3.1 Satellite observations and CMIP5/6 simulations**

The first two columns in Fig. 2 (from a to e) show the estimated long-term mean cloud albedo corresponding to the correlation between planetary albedo and cloud fraction over the five regions from the observation and 22 AMIP5/6 models, including 10 individual models and an ensemble mean for AMIP5 and AMIP6 (represented by AMIP5-MEM and AMIP6-MEM), during the period from 2003 to 2008. For the combined satellite observations, the correlation coefficient values are 210 above 0.85 in all regions. The correlation over the Peruvian region was the largest (~0.95), while a relatively weak correlation (~0.88) appeared in the Canarian region. Such a high correlation between planetary albedo and cloud fraction further indicates the homogeneity of cloud and surface types over these regions. The regressed cloud albedo from the satellite ranged from 0.30 to 0.42 for the five stratocumulus regions, which is consistent with previous studies (Bender et al., 2011; Engström et al., 2014). As the values averaged over Aqua and Terra albedos and cloud fractions were used as the 215 observation in this study, the regressed cloud albedo values need to be within the range of the Aqua and Terra (Engström et al., 2014). Regarding the AMIP5 and AMIP6 models, a higher correlation (> 0.8) appeared for most models at the five regions, especially higher at the Australian and Canarian regions. At the Peruvian, Namibian and Californian regions, the correlations of the observation were relatively higher than those of most climate models while the observed correlation was approximately close to the median value of model simulations at the Australian and Canarian regions.

Although previous studies indicated that some CMIP6 models updated the cloud physical parameterization in the new version (Seland et al., 2020; Kawai et al., 2020), the correlation coefficients of the AMIP6 models between planetary albedo and cloud fraction showed a lower value than those of the AMIP5, indicating that the linear relationship between cloud fraction and planetary albedo in the AMIP6 models' simulations is not superior to that of AMIP5. While the AMIP6 simulations displayed a similar spread in the estimated cloud albedo for all regions, some AMIP6 models produced a lower correlation coefficient than those of the AMIP5 models (e.g., AMIP6/INM-CM4-8). Notably, the AMIP5-MEM and AMIP6-MEM always produced a worse correlation relationship and more irrational cloud albedo values, indicating that the AMIP5/AMIP6 models have a lack of skill in simulating cloud properties over the marine stratocumulus regions.

The third and fourth columns in Fig. 2 (from f to j) also show the estimated long-term mean cloud albedo and the correlation between planetary albedo and cloud fraction over the five regions for the observation and 29 AMIP6 models during 2003 to 2014. The simulated correlation exhibited a larger spread at the Peruvian and Namibian regions than those at other regions, indicating that the AMIP6 models have a lack of capacity to capture the linear relationship between planetary albedo and cloud fraction. The cloud albedos were underestimated in most CMIP6/AMIP models compared with the satellite-based cloud albedos. The Australian (0.30~0.43) and Canarian (0.24~0.42) regions displayed a larger inter-model variability in the cloud albedo than other regions due to a poor skill in simulating the cloud properties (e.g., LWP and COT). Over the Canarian regions, the correlation and cloud albedo of AMIP6-MEM showed good agreement with those of the satellite observation compared with those of the individual AMIP6 models, resulting from the offsetting effect between models. Overall, the AMIP6 models reproduced the cloud albedo and correlation well at the Australian region while a higher uncertainty in model's simulations, i.e., a larger intermodal spread, at the Peruvian region (Engström et al., 2014).

Engström et al. (2014) also found that CMIP5 models simulating a higher cloud cover have a tendency to produce a smaller cloud albedo value. Darker clouds can offset the contribution of the higher cloud cover to the planetary albedo, resulting in relatively a consistent model-driven planetary albedo. This is a presentation of the "too few, too bright" problem that persists in GCMs (Nam et al., 2012). To validate whether or not this problem has been improved in the AMIP6 models, we compared the relationship between regressed cloud albedo and cloud fraction (See Fig. S1). The correlations driven by the 28 AMIP6 models were -0.28, 0.19, -0.11, -0.71 and 0.43 for the Peruvian, Namibian, Californian, Australian and Canarian regions, respectively. Compared with the results from the CMIP5 models (Engström et al., 2014), noticeable progress was found at the Namibian and Californian regions while a high negative correlation was simulated at the Australian region, indicating that the new generation models need to be further improved to resolve the longstanding problem.

The monthly cloud albedo time series regressed from the satellite, MEM and AMIP6-MEM for the six-year period from 2003 to 2008 over the five regions are shown in Fig. 3(a-e). The temporal correlations (R5/R6) and corresponding confidence value (P5/P6) between simulated (AMIP5-MEM/AMIP6-MEM) and satellite regressed monthly cloud albedo time series are also given in Fig. 3(a-e). Note that the smoothed time series were produced by 12-month smoothing. The statistical results showed that the R5/R6 values were 0.62/0.78, 0.44/0.55, 0.38/0.45, 0.75/0.74 and 0.00/0.05 for the Peruvian, Namibian, Californian, Australian and Canarian regions, respectively. Among them, the correlations only at the

Canarian region were insignificant (i.e., P5/P6=1.00/0.70). A high positive correlation appeared at the Australian region (>0.70), indicating that the changes in the cloud albedo are well captured by the models.

Compared with AMIP5-MEM, the regressed monthly cloud albedo of AMIP6-MEM showed a better correlation with the satellite regressed values. However, the performance of AMIP5-MEM in reproducing monthly cloud albedo and its amplitude (the difference between the maximum and minimum values of cloud albedo) was better than that of AMIP6-MEM. Furthermore, the monthly cloud albedos obtained from the satellite and models displayed obvious seasonal cycle at all regions except for the Canarian region. This may be related to the fact that a weaker linear relationship between monthly cloud cover and planetary albedo may exist at the Canarian region, resulting in a significant change in the estimated cloud albedo (see Fig. S2).

In addition, the monthly cloud albedo time series for the satellite and AMIP6-MEM for the period from 2003 to 2014 in the five regions are also shown in Fig. 3(f-j), which are consistent with Fig. 3(a-e), indicating that the simulation capability of the AMIP6-MEM in different regions is not improve significantly with the expansion of the simulation time and the increase of the model numbers. The amplitudes of the cloud albedo simulated from the model were larger than that of the satellite at the Peruvian, Namibian and Californian regions while smaller at the Australian and Canarian regions. Note that at the Australian region, the monthly cloud albedo exhibited a large variation than that at other regions based on the satellite-based observation, which means that the cloud optical properties (e.g., COT and cloud effective radius) have been considerably changed within the Australian region.

This study further assessed the performance of the AMIP6 models in reproducing the cloud albedo time series. Figs. 4a-e provide the Taylor diagrams (Taylor, 2001) for the five regions, which include the correlation coefficients, the centered root mean square error (RMSE, the green circle), and the standard deviation value between individual AMIP6 models and the satellite-based observations. The centered RMSE and the standard deviation values represent the model's ability to reproduce the phase and amplitude of the variable, respectively. Correlation coefficients greatly varied with regions, ranging from negative (Peruvian, Namibian and Canarian) and positive values. Compared with other regions, most of the models showed a high positive correlation (>0.6) at the Peruvian region. The model-driven cloud albedo was most poorly correlated with the observation at the Canarian region, e.g., < 0.4 or negative values. On the contrary, at the Australia region, all models showed a significant positive correlation (> 0.4). The standard deviation values of the models at the Peruvian, Namibian and California ranged 0.02-0.09, 0.02-0.11 and 0.03-0.10, respectively while 0.03 for the satellite-based observation. This result indicates that most of the models overestimate the amplitude of the cloud albedo time series at the regions. Some models produced the standard deviation values of cloud albedos three times larger than the observation. It is evident that the standard deviation values of the simulated cloud albedo at the Australian regions were closer to the observed value than that of other regions, indicating that the AMIP6 models also perform well in simulating the amplitude of the monthly cloud albedo time series at this region. Overall, the intermodal variability in the correlation coefficient, RMSE and standard deviation values was the smallest at the Australian region while the largest at the Peruvian region.

Further, Figure 5 shows the annual cycles of the cloud albedo estimated by the satellite and AMIP5/AMIP6 models for the five regions. The seasonal variation in the cloud albedo at each region takes a shape of single peak distribution. In terms of similarity among regions, the cloud albedo at all regions reached the maximum value during the boreal winter season, i.e.,
December to January in the Northern Hemisphere while June to July in the Southern Hemisphere. Many previous studies have demonstrated that the seasonal variations of marine cloud properties (e.g., cloud fraction, LWP and cloud thickness) are strongly affected by meteorological conditions(Lin et al., 2009; Wood, 2012; Dong et al., 2014). Employing a 19-month record of ground-based lidar/radar observations from the Atmospheric Radiation Measurement Program Azores site, for example, Dong et al. (2014) found that the seasonal variations of cloud thickness and LWP are closely related to the seasonal
synoptic patterns (e.g., transport of water vapor, relative humidity, high/low pressure system). Furthermore, the influence of aerosols loading is non-neglectable. While the aerosols act as CCN, the concentration of CCN can significantly influence the cloud albedo of low clouds (Twomey, 1974). On the other hand, absorbing aerosols near stratocumulus may enhance absorbing solar energy, resulting in an influence on the dynamical evolution of stratocumulus causing a change in the cloud albedo (Wilcox, 2010). The seasonal cycle of the cloud albedo at the Australia region showed the largest amplitude among
the five regions (ranging from 0.37 to 0.52) while the amplitudes at other regions were less than 0.10. Such a result means that the meteorological conditions and aerosol loadings of the cloud system at the Australian region have a relatively larger seasonal variation compared with those at other regions.

The COT usually increases with an increase in cloud LWP, resulting in an increase in the cloud albedo (Wood, 2012). Gryspeerdt et al. (2019) also concluded that LWP is the main factor controlling liquid cloud albedo. Thus, this study
investigated the seasonal variation of LWP and found that the change in LWP is strongly correlated with the change in cloud albedo at the Peruvian, Australian and Canarian regions (see Fig. S3). For the Namibian region, however, many studies have shown that the continuous transportation of absorbing biomass burning aerosols from Africa to the region during the African biomass burning season from August to October (Das et al., 2017) can reside above the clouds, resulting in an increase in the cloud albedo by thickening the stratocumulus (Wilcox, 2010, 2012). Zuidema et al. (2018) also found that the biomass
burning aerosols generally exist in the boundary layer at the earlier time of the biomass burning seasons and are mainly located at above the clouds in September to October, which is caused by the northwestward transportation of the biomass burning aerosols from the African continent. However, Fig. 5b shows that the peak of the cloud albedo occurred in July and then continuously decreased from August to October at the Namibian, indicating that the changes in the cloud albedo are difficult to be explained by the negative semi-direct effect of the biomass burning aerosols. This result is consistent with the
work of Bender et al. (2016) which concluded that the direct effect and positive semi-direct effect are the main aerosol effects (Wilcox, 2012). That is, clouds become darker under a polluted environment. Regarding the seasonal cycles of cloud droplet number concentration $N_d$ , we found that the seasonal cycles of the cloud albedo at the Namibian region were highly correlated with those of $N_d$ obtained from The Cloud-Aerosol Lidar and Infrared Pathfinder Satellite Observations (CALIPSO) (Li et al., 2018), whereas the seasonal cycles of $N_d$ and the cloud albedo showed opposite seasonal changes to
each other at the California region. The relationship between the $N_d$ and the cloud albedo varies with different regions, which

may be caused by the effect of meteorological conditions. These results indicate that it is a challenge to study the variability in the cloud albedo over the marine stratocumulus regions under various meteorological and aerosol conditions.

Fig. 5(a-e) shows the seasonal cycles of cloud albedo at the five regions during a period from 2003 to 2008 for the AMIP5/AMIP6 and the satellite-based observation. Shading areas in Fig. 5 represent the range of the cloud albedo simulated by the 22 models. The R5/R6 and P5/P6 values for the seasonal cycles of the cloud albedo obtained from the models and the satellite-based observation are also given in Fig. 5. For the AMIP5-MEM and AMIP6-MEM, the correlations of the cloud albedo seasonal cycles between the models and the observation are highly positive at all regions (R5/R6>0.6), except for the Canarian region (R5/R6=0.22/0.53). The R values were the largest at the Namibian (R5/R6=0.82/0.92) and Australian regions (R5/R6=0.93/0.92). Overall, the results of AMIP6 were slightly superior to those of AMIP5, especially at the Canarian region. However, the seasonal cycles of cloud albedo estimated from the AMIP6-MEM at the Canarian region for 12 years from 2003 to 2014 (Fig. 5j) exhibited a significant negative correlation with that of the satellite-based observation, indicating AMIP6-MEM still has a lack of skill to capture the seasonal cycle of the cloud albedo at this region even if the numbers of AMIP6 models increases.

## 3.2 The impacts of different aerosol types and meteorological factors on cloud albedo changes

Cloud liquid water may affect the COT, which is subsequently influencing the cloud albedo (Wood, 2012). Furthermore, the change of LWP also may influence the relationship between aerosols and cloud properties (Robert et al., 2008; Gryspeerdt et al., 2019; Douglas and L'Ecuyer, 2019). For example, the effect of aerosols on the cloud albedo may be weakened by a change in the LWP (Han et al., 2002; Twohy, 2005). Based on in situ observations, recent studies found that the relationship between aerosol concentration and cloud droplet effective radius changes from negative to positive when liquid water content increases (Qiu et al., 2017; Zhao et al., 2019b). Considering the effect of LWP, this study evaluated the impact of meteorological parameters and aerosol types on the cloud albedo at different LWP ranges in order to evaluate the influence of LWP on cloud albedo. Firstly, the 720 monthly sample data obtained from the five regions were divided into two groups based on the range of monthly mean LWP values: LWP $\leq$ 65 g m$^{-2}$ and 65 g m$^{-2}$ < LWP $\leq$ 120 g m$^{-2}$. Here, the threshold of 65 g m$^{-2}$ for LWP was chosen to evenly split the samples.

Figure 6a-b shows the regression coefficients in the partial correlation calculation and the relative contributions for individual variables related to cloud albedo changes under different LWP conditions. Normalized variables were incorporated into the regression models. There is a considerable discrepancy in the results between the two groups. For the lower LWP bin (i.e., LWP $\leq$ 65 g m$^{-2}$), the results showed that the regression coefficient related BC/SO2/SS to the cloud albedo was positive while DU and OC-related coefficients were negative, which indicates that the cloud albedo increases with increasing BC/SO2/SS and decreases with increasing DU/OC. Fig. 6b also clearly shows that DU, BC and OC have a larger contribution to the change in the cloud albedo compared with other predictors, e.g., omega900, EIS and RH700. Under

LWP > 65 g m$^{-2}$, the contribution of DU to the cloud albedo was the largest. In addition, SO2 and SO4 also considerably contributed to the cloud albedo

In addition to the effects of LWP, the difference in the relative contribution may be induced by the regional variability in
aerosol types. A smaller LWP mainly appeared at the Namibian and Canarian where the main aerosol types are DU and BC, while lower BC loadings were found at the regions with a larger LWP (Fig. S4). While the positive coefficient for BC reflects the indirect effect of aerosols on the cloud albedo, the negative dependency of BC may represent the direct and semi-direct effects of absorbing aerosols (Johnson et al., 2004; Bender et al., 2016). For example, Johnson et al. (2004) found that absorbing aerosols in clouds can make the clouds warmer and thinner, resulting in a decrease in cloud albedo. Besides,
McCoy et al. (2018) found a negative dependence of $N_d$ on BC at regions with low BC loadings. This means that a decrease in the cloud albedo may be associated with a decrease in $N_d$. The dependence of $N_d$ on OC has been also investigated in previous studies (McCoy et al., 2018; Li et al., 2018) and a negative dependence of $N_d$ on OC has been found in some marine regions. The negative sensitivities of OC to the cloud albedo may attribute to a decrease of $N_d$ with an increase of OC.

Dust is a crucial predictor of the cloud albedo and the coefficient of DU was negative for the two datasets divided in this
study, which may be induced by the semi-direct effects of absorbing aerosols. In literature, many studies have examined the impacts of dust aerosols on stratocumulus (Doherty and Evan, 2014; Amiri-Farahani et al., 2017). For example, Karydis et al. (2011) showed that aged dust reduces $N_d$ by consuming the supersaturation of clouds. However, Mccoy et al. (2017) estimated the indirect effect of aerosol from satellite observations and reanalysis data and found that the dust has a limited impact on $N_d$ in different stratocumulus regions. Pradelle et al. (2002) employing satellite observations also investigated the
effect of Saharan dust on marine stratocumulus clouds and found that minimum cloud albedo values appeared in regions with the most dust particles. They also found that the dust in a stratiform cloud may decrease the initial CCN and increase the effective droplet radius, which causes reducing the cloud albedo (Pradelle and Cautenet 2002). In addition, a recent study also showed that the dust aerosol can even further influence the meteorological environment that the clouds form by both suppressing the SST and affecting the temperature and humidity profile (Sun et al., 2020). A significant influence of dust on
the cloud albedo in this study may be driven by the collected samples at the five regions where the cloud albedo and dust highly vary with the regions.

Under LWP ≤ 65 g m$^{-2}$, the coefficient of SS was a small positive value while the correlation coefficient of sea salt was insignificant under LWP > 65 g m$^{-2}$, which means that these variables are not suitable as a predictor for estimating the cloud albedo. This is consistent with the results of McCoy et al. (2017; 2018) which indicate that the $N_d$ is weakly dependent on
the SS although sea salt is an effective CCN. McCoy et al. (2018) have also validated the influence of SS on $N_d$ with up-to-date observations. As submicron SS in the MERRA-2 reanalysis data can be simply predicted from wind speed and SST by a parameterization (Jaeglé et al., 2011; McCoy et al., 2018; Li et al., 2018), the effect of SS on the cloud albedo may be dependent on the relationship between the cloud albedo and near-surface wind speed, which may explain the limited effect of SS on the cloud albedo.

The coefficients of SO2 were positive for both datasets. In addition, the Twomey effect for SO2 was further pronounced under the condition with higher LWP. The previous studies (e.g., McCoy et al., 2017; Li et al., 2018) showed that SO4 plays a key role in modulating $N_d$. Although their results showed significant positive coefficients of SO4 with $N_d$, this study found an unexpected negative correlation of SO4 with the cloud albedo. Such a result may be driven by the fact that the sulfate aerosol particles and dust are externally mixed. The previous studies showed that sulfate-covered dust can act as CCN, which

may induce a decrease in the cloud albedo by enhancing the collision-coalescence progress of droplets (Levin et al., 1996; Rosenfeld et al., 2001).

The results of this study showed a weak dependency of the cloud albedo with omega900, RH700 and EIS. Under LWP ≤ 65 g m$^{-2}$, the upward vertical velocity and RH700 have an unexpected negative but weak effect on the cloud albedo and the relative contributions of omega900 and RH700 are negligible. Under LWP > 65 g m$^{-2}$, no significant correlation between the

cloud albedo and omega900 was found. Note that the analysis of this study employed the average data at the monthly scale rather than raw satellite measurements at an instantaneous scale, which may make the cloud albedo less sensitive to omega900. The coefficient of RH700 was positive and the relative contribution was about 4%. Generally, drier free-troposphere humidity usually drives stronger entrainment of dry air, which induces evaporating and raising lifted condensation level, resulting in a reduced cloud thickness (Wood, 2012; Eastman and Wood, 2018). The positive

dependency of the cloud albedo with EIS was identified for the two datasets divided in this study, which may be caused by stronger inversions linked to increased stability and reduced vertical exchange at cloud top, resulting in thicker low clouds by keeping moisture trapped in the marine boundary layer (MBL) (Scott et al., 2020). Compared to other meteorological factors, the contribution of SSTadv to cloud albedo was larger, non-negligible in both datasets (7%~9%). Under LWP ≤ 65 g m$^{-2}$, the SSTadv showed a negative coefficient. The cold advection usually thickens clouds by reducing low level stability

and transporting more moisture into the MBL (George and Wood, 2010; Scott et al., 2020). Under LWP > 65 g m$^{-2}$, the coefficient of SSTadv was positive, which is hard to be explained by the aforementioned mechanism. By analysing the correlation between LWP and SSTadv, we found that there was no significant correlation between them. This indicates that the surface temperature advection may affect cloud albedo in other ways than by affecting the moisture in cloud layers. Furthermore, dust can affect the meteorological environment through radiative effects, consequently, the positive coefficient

found in this study may be a reflection of their effects (Sun et al., 2020; Huang et al., 2021). The coefficients of the omega700 were negative for both datasets. The downdraft allowed dry air above the cloud to enter the clouds, causing evaporation and making cloud droplets smaller and less, resulting in reducing the cloud albedo (Yang et al., 2019). Note that the role of omega700 was very weak under the condition with higher LWP, and its contribution was negligible.

The analysis of the relative contribution of each predictor variable was similar to the results of the coefficients. Under LWP

≤ 65 g m$^{-2}$, DU and BC contributed approximately 63 % variations of the cloud albedo in the regression model. Note that the contribution of omega700 and SSTadv was non-negligible, accounting for 18 %. Under LWP > 65 g m$^{-2}$, the contribution of DU and SO2 to the change of cloud albedo was about 61 %. DU has the largest relative contribution to the cloud albedo changes (~35 %) in both datasets.

The normalized satellite-based and model-driven cloud albedos under different cloud water conditions are shown in Figure
6c-d, where the correlation (R) between the two cloud albedos is given in parentheses. A larger R value indicates a better
model. Both of the correlation coefficients are greater than 0.65, indicating the regression model properly captures the
changes in the cloud albedo for the two datasets. A considerable part of the variation in cloud albedo can be explained by the
change in meteorological parameters and mass concentrations of different aerosol types.

In addition, to verify the sensitive of the results to input data, we employ different datasets to perform the multi-linear
regression. The monthly Multisensor Advanced Climatology of Liquid Water Path (MAC-LWP) is used to test the sensitive
of the results to input LWP data (Elsaesser et al., 2017). Considering the differences in retrieval methods and values of the
MODIS LWP and MAC-LWP datasets (Greenwald, 2009), we used the threshold of 55 g m$^{-2}$ for MAC-LWP to better split
the samples evenly. The regressed results are given in Figure S5. We can see that the results did not change significantly,
indicating that the regressed results are relatively robust. For the reanalyzed dataset, ERA-5 reanalysis is considered to be the
most state-of-the-art reanalysis with higher temporal and spatial resolutions (Hersbach et al., 2019). We also used the ERA-5
data to perform the multiple regression model (see Figure S6). Although the results change slightly, the changed results do
not affect the main conclusions.

It is also found from Figure 6 that changes in LWP can also cause an alteration of the relationship between aerosol and the
cloud albedo. To further investigate the influence of meteorological factors on the relationship, the partial correlations were
calculated to eliminate the influence of meteorological parameters individually or simultaneously. If the partial correlation is
similar to the total correlation, it means that the influence of meteorological factors on the relationship is limited. In contrast,
the influence of meteorological factors on the relationship may be significant if the partial correlation and the total
correlation are the opposite sign. Given six meteorological parameters (omega700, omega900, RH700, EIS, SSTadv and
LWP) considered in this study, the total correlation and partial correlation between the cloud albedo and different aerosols
for two sample groups are given in Table 3. Under LWP $\leq$ 65 g m$^{-2}$, the correlations of all aerosol types were weakened
when eliminating the effects of meteorological factors. When the influence of EIS and LWP were eliminated, the correlation
of DU becomes much weaker, indicating that the correlation of DU is sensitive to EIS and LWP. On the contrary, the
correlations of BC, OC and SO4 were stronger when the influence of LWP was eliminated. In addition, most aerosol types
were sensitive to SSTadv except for the SS. The correlation of BC/OC ranged from 0.21/0.20 to -0.03/-0.05 by eliminating
the influence of SSTadv, indicating that the relationship between BC/OC and the cloud albedo is extremely sensitive to the
influences of SSTadv. Under LWP > 65 g m$^{-2}$, the correlations of all aerosol types varied significantly by eliminating the
influence of all meteorological parameters. E. g., the correlation of BC/DU/OC ranged from -0.47/-0.49/-0.45 to -0.01/-
0.02/-0.03. This indicates that the cloud-aerosol interaction is more sensitive to the response of meteorological conditions at
higher LWP conditions. Although the contribution of meteorological parameters to the change in the cloud albedo is only a
small part based on relative contribution calculation, its influence on cloud-aerosol interactions is not negligible.

## 4 Conclusions and discussion

The cloud albedo at the marine subtropical stratocumulus regions has a key role in regulating the regional energy budget. However, climate models have a lack of skill to properly capture the cloud properties over the regions. Therefore, the CMIP6 has more attention to improve some long-standing model biases, e.g., the low cloud simulation over tropical oceans and surface processes (Stouffer et al., 2016). Accordingly, considerable improvements in reproducing the observed seasonal planetary albedo over the subtropical stratocumulus have been found in CMIP6 (Jian et al., 2020). To enhance the confidence in climate predictions, it is necessary to systematically evaluate and compare the performance of CMIP5 and CMIP6 models and to further study the processes that contribute to the cloud albedo using the satellite-driven and reanalysis data. This study investigated the performances of CMIP6 models in reproducing the cloud albedo at the five marine subtropical stratocumulus regions from 2003 to 2014.

For the long-term regressed values, the cloud albedos were underestimated in most AMIP6 models compared with the satellite-driven cloud albedos. The AMIP6 models produced a similar spread of AMIP5 at all regions, even some AMIP6 models performed worse than AMIP5. The monthly cloud albedo of AMIP6-MEM showed better correlation with the satellite-driven observation than that of AMIP5-MEM. However, this study found a lack of skill in reproducing the values and amplitude at some regions (e.g., Peruvian and Namibian), indicating that the cloud parameterization between two generations of AMIP models needs to be further improved to produce more accurate predictions. This study also found that most AMIP6 models overestimated the amplitude of the cloud albedo at all regions except for the Australian region, i.e., simulating higher seasonal variations. Overall, the AMIP6 models performed the best at the Australian region and the worst at the Canarian region. The seasonal cycle of cloud albedo of AMIP6-MEM was correlated better with satellite-driven observations than that of AMIP5-MEM. For the Australian region, the model-driven seasonal cycle of the cloud albedo was almost consistent with that of the satellite-driven observation, which indicates the superiority of model performance at this region.

Employing the satellite and reanalysis data, we further evaluated the impacts of different aerosol types and meteorological factors on the cloud albedo. Changes in aerosol types and meteorological factors explained ~65 % of the changes in the cloud albedo. However, the controlling factors and their contribution rates varied with LWP conditions. Under the monthly mean LWP $\leq$ 65 g m$^{-2}$, DU and BC dominantly contributed to the changes in the cloud albedo, while DU and SO2 contributed the most under the condition of 65 g m$^{-2}$ < LWP $\leq$ 120 g m$^{-2}$. Although the contributions of aerosols were significant, the influence of meteorological factors on the cloud-aerosol interactions cannot be ignored.

Due to the limitations of polar-orbiting satellite observations, this study did not obtain a complete diurnal cycle of cloud properties and radiation flux, which may induce a bias in the results of this study. The diurnal cycle of marine subtropical stratocumulus cloud albedo is usually significant due to the diurnal cycle of solar energy (Wood, 2012). The maximum cloud thickness usually occurs in the morning and gradually decreasing over the afternoon due to absorbing solar radiation in the cloud layer (Wood et al., 2002; Christensen et al., 2013). It is a challenge to evaluate how much of the cloud albedo bias

contributes to the diurnal cycle of cloud albedo. Therefore, it is necessary to evaluate the diurnal cycle of cloud albedo in the

marine subtropical stratocumulus regions for reducing the uncertainties in cloud radiation interactions in GCMs. Note that the "too bright, too few" problem was improved at the Namibian and Californian regions in AMIP6. However, even if some models can simulate the cloud albedo more reasonably, it is questionable if other cloud properties can be captured (e.g., total cloud fraction), consequently resulting in significant biases in radiation (see Fig. S7). Therefore, we need to pay more attention to improve the calculation of total cloud fraction in the GCMs. Recently, some studies are devoted to improve

cloud overlap parameterization for accurately simulating the cloud fractions in GCMs (Li et al., 2018, 2019). Accordingly, it is also necessary to evaluate the improvement of cloud overlap scheme on cloud radiation interaction using long-term satellite-driven observations and reanalysis data.


**Data availability**

The CERES datasets are available from the CERES website: https://ceres.larc.nasa.gov/data/#single-scanner-footprint-ssf. The MODIS datasets are available from the  Level-1 and Atmosphere Archive & Distribution System (LAADS) Distributed Active Archive Center (DAAC) website: https://ladsweb.modaps.eosdis.nasa.gov/archive/allData/61. The MERRA-2

reanalysis products are downloaded from the MERRA-2 website: https://disc.gsfc.nasa.gov/datasets?keywords=MERRA-2. The CMIP5 and CMIP6 products are downloaded from the Earth System Grid Federation (ESGF)  website: https://esgf-node.llnl.gov/projects/esgf-llnl/.

**Competing interests**

The authors declare that they have no conflict of interest.

**Author contribution**

BJ and JL organized the paper and carried them out. BJ prepared the manuscript with contributions from all co-authors. JL conceptualized the paper and revised the whole manuscript. GW provided the computing resources. YL downloaded the data

and maintain research data. YZ, JW and MZ processed the raw model output data into consistent gridded format for comparison with the satellite dataset. JH provided consultations and acquired the financial support for this study. All authors contributed to the discussion of the results and reviewed the manuscript.

## Acknowledgments

This research was jointly supported by the Strategic Priority Research Program of the Chinese Academy of Sciences (XDA2006010301), the National Science Fund for Excellent Young Scholars (42022037). The authors declare that they have no conflict of interest. We would like to thank the CERES, MODIS, CMIP5, CMIP6 and MERRA-2 science teams for providing excellent and accessible data products that made this study possible.

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

**Table 1: The list of CMIP5 models used in the study and their atmospheric horizontal resolutions.**

| Model name | Origin | Resolution (lon×lat) |
| --- | --- | --- |

| | | | |
|---|---|---|---|
| 1 | ACCESS1-0 | Commonwealth Scientific and Industrial Research Organization and Bureau of Meteorology, Australia | 192×145 |
| 2 | ACCESS1-3 | | 192×145 |
| 3 | FGOALS-g2 | Institute of Atmospheric Physics, Chinese Academy of Sciences and Tsinghua University, China | 128×60 |
| 4 | GISS-E2-R | NASA/Goddard Institute for Space Studies, USA | 144×90 |
| 5 | INMCM4 | Institute for Numerical Mathematics, Russia | 180×120 |
| 6 | IPSL-CM5A-LR | Institut Pierre Simon Laplace, France | 96×96 |
| 7 | MIROC5 | AORI, NIES and JAMSTEC, Japan | 256×128 |
| 8 | MPI-ESM-LR | Max Planck Institute for Meteorology, Germany | 192×96 |
| 9 | MRI-CGCM3 | Meteorological Research Institute, Japan | 320×160 |
| 10 | NorESM1-M | Norwegian Climate Centre, Norway | 144×96 |

**Table 2: The list of CMIP6 models used in the study and their atmospheric horizontal resolutions.**

| | Model name | Origin | Resolution (lon×lat) |
|---|---|---|---|
| 1 | ACCESS-CM2 | Commonwealth Scientific and Industrial Research Organization and Bureau of Meteorology, Australia | 192×145 |
| 2 | ACCESS-ESM1-5 | | 192×145 |
| 3 | FGOALS-g3 | Institute of Atmospheric Physics, Chinese Academy of Sciences and Tsinghua University, China | 180×80 |
| 4 | GISS-E2-1-G | NASA/Goddard Institute for Space Studies, USA | 144×90 |
| 5 | INM-CM4-8 | Institute for Numerical Mathematics, Russia | 180×120 |
| 6 | IPSL-CM6A-LR | Institut Pierre Simon Laplace, France | 144×143 |
| 7 | MIROC6 | AORI, NIES and JAMSTEC, Japan | 256×128 |
| 8 | MPI-ESM1-2-HR | Max Planck Institute for Meteorology, Germany | 384×192 |

| 9 | MRI-ESM2-0 | Meteorological Research Institute, Japan | 320×160 |
|---|---|---|---|
| 10 | NorESM2-LM | Norwegian Climate Centre, Norway | 144×96 |
| 11 | BCC-CSM2-MR | Beijing Climate Center, China | 320×160 |
| 12 | BCC-ESM1 | | 128×64 |
| 13 | CAMS-CSM1-0 | Chinese Academy of Meteorological Sciences , China | 320×160 |
| 14 | CESM-FV2 | | 144×96 |
| 15 | CESM2-WACCM | National Center for Atmospheric Research, Climate and Global Dynamics Laboratory, USA | 288×192 |
| 16 | CESM2 | | 288×192 |
| 17 | CESM2-WACCM-FV2 | | 144×96 |
| 18 | CanESM5 | Canadian Centre for Climate Modelling and Analysis, Environment and Climate Change Canada, Canada | 128×64 |
| 19 | E3SM-1-0 | LLNL, ANL, BNL, LANL, LBNL, ORNL, PNNL and SNL, USA | 360×180 |
| 20 | EC-Earth3-Veg | EC-Earth consortium (27 institutions in Europe) | 512×256 |
| 21 | EC-Earth3 | | 512×256 |
| 22 | FGOALS-f3-L | Chinese Academy of Sciences, China | 288×180 |
| 23 | INM-CM5-0 | Institute for Numerical Mathematics, Russian Academy of Science, Russia | 180×120 |
| 24 | KACE-1-0-G | National Institute of Meteorological Sciences/Korea Meteorological Administration, Republic of Korea | 192×144 |
| 25 | NESM3 | Nanjing University of Information Science and Technology, China | 192×96 |
| 26 | NorCPM1 | NorESM Climate modeling Consortium consisting of CICERO, MET-Norway, NERSC, NILU), UIB, UIO and UNI, Norway | 144×96 |
| 27 | SAM0-UNICON | Seoul National University, Republic of Korea | 288×192 |
| 28 | TaiESM1 | Research Center for Environmental Changes, Academia Sinica, Taiwan | 288×192 |

**Table 3: Total correlations between the cloud albedo and different aerosol types, and the partial correlations to eliminate the influence of three meteorological parameters individually or simultaneously under different LWP conditions. The value above is under the condition of LWP ≤ 65 g m$^{-2}$. The value in parentheses is under the condition of 65 g m$^{-2}$ < LWP ≤ 120 g m$^{-2}$.**

|  | BC | DU | OC | SO2 | SO4 | SS |
|---|---|---|---|---|---|---|
| Total correlation | 0.21 | -0.51 | 0.20 | 0.32 | 0.36 | -0.29 |
|  | (-0.47) | (-0.49) | (-0.45) | (-0.10) | (-0.55) | (0.03) |
| Omega700 | 0.18 | -0.46 | 0.18 | 0.27 | 0.32 | -0.25 |
|  | (-0.45) | (-0.51) | (-0.43) | (-0.11) | (-0.54) | (0.09) |
| Omega900 | 0.20 | -0.48 | 0.20 | 0.30 | 0.37 | -0.25 |
|  | (-0.50) | (-0.54) | (-0.48) | (-0.12) | (-0.55) | (0.03) |
| RH700 | 0.21 | -0.45 | 0.22 | 0.25 | 0.36 | -0.23 |
|  | (-0.46) | (-0.50) | (-0.44) | (-0.06) | (-0.55) | (0.01) |
| EIS | 0.14 | -0.38 | 0.14 | 0.29 | 0.18 | -0.14 |
|  | (-0.30) | (-0.43) | (-0.26) | (-0.20) | (-0.33) | (0.03) |
| SSTadv | -0.03 | -0.43 | -0.05 | 0.13 | 0.13 | -0.29 |
|  | (-0.29) | (-0.36) | (-0.27) | (0.05) | (-0.37) | (0.17) |
| LWP | 0.33 | -0.31 | 0.30 | 0.42 | 0.34 | -0.15 |
|  | (-0.40) | (-0.35) | (-0.39) | (0.09) | (-0.53) | (-0.08) |
| All parameters | 0.14 | -0.23 | 0.11 | 0.24 | 0.20 | -0.15 |
|  | (-0.01) | (-0.02) | (-0.03) | (0.12) | (0.05) | (-0.07) |

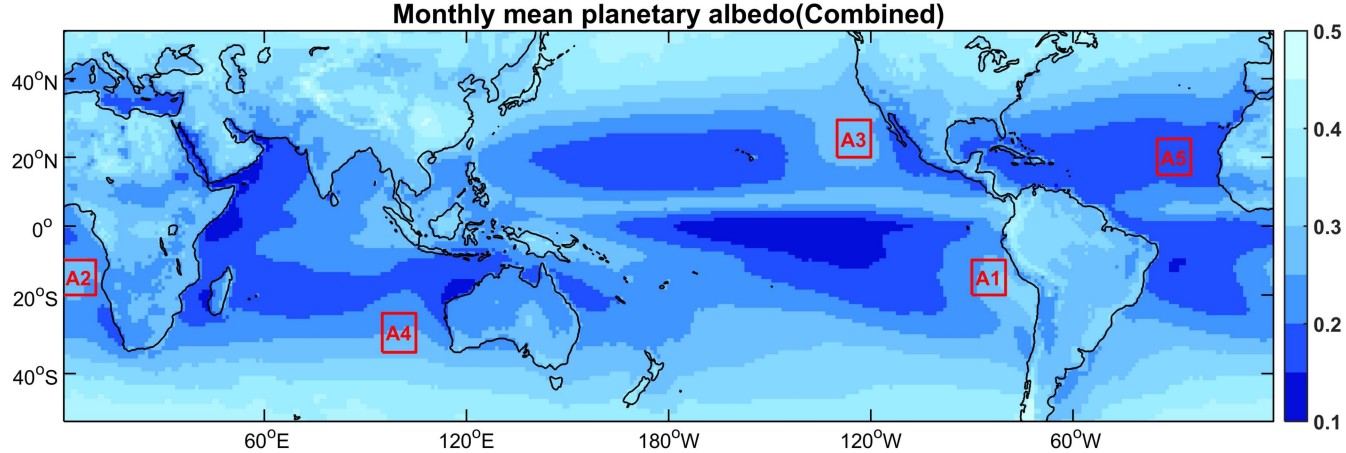

**Figure 1: Near-global distribution of combined planetary albedo averaged from Aqua and Terra during 2003-2014. Red rectangular boxes indicate the five regions used chosen for the analysis: (A1) Peruvian, (A2) Namibian (A3) Californian, (A4) Australian and (A5) Canarian.**

790

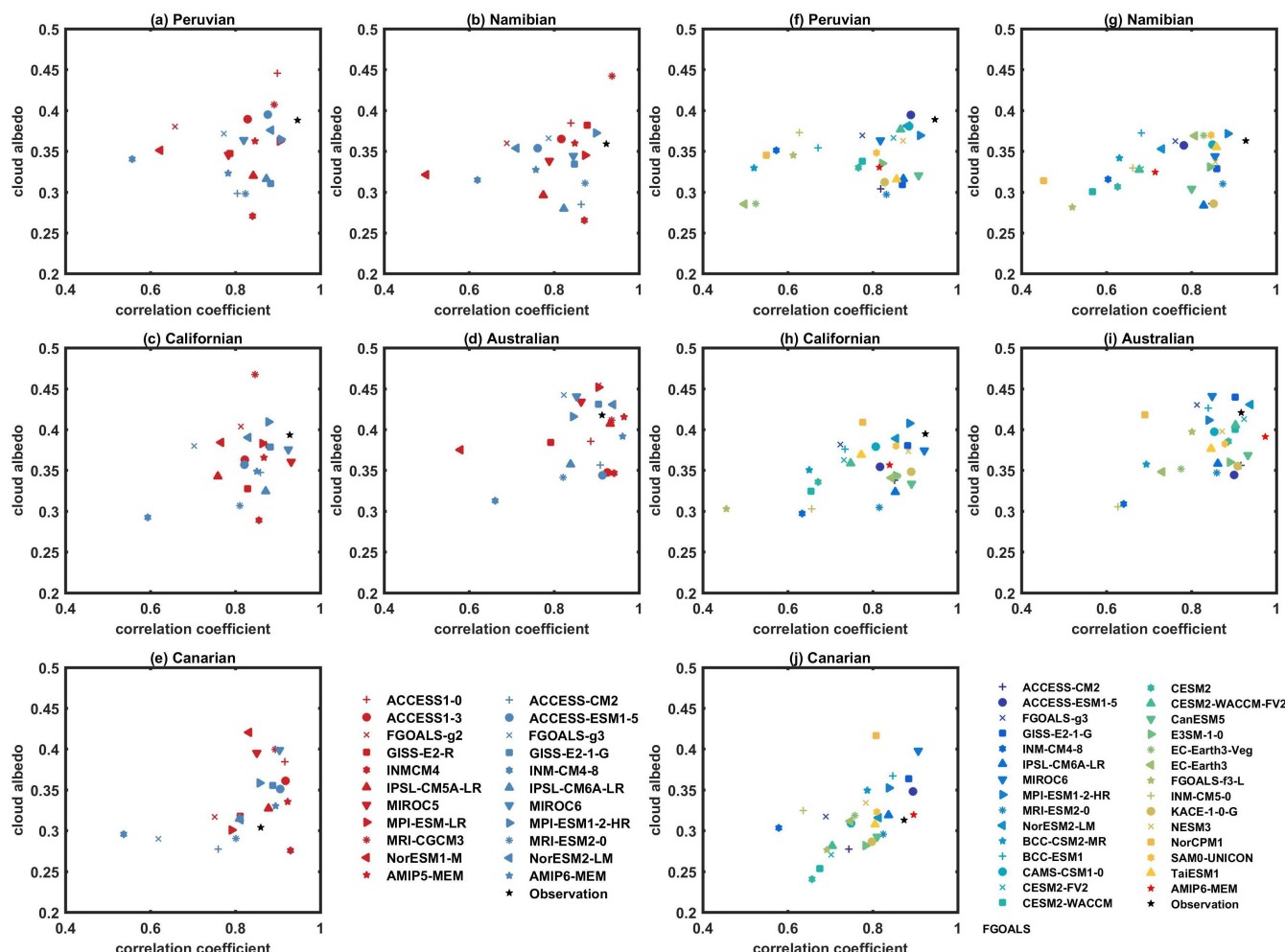

**Figure 2:** The estimated long-term mean cloud albedo and corresponding correlation coefficient from the relation between planetary albedo and cloud fraction (a-e) from satellite observations (black symbol), 11 AMIP5 (red symbols) and 11 AMIP6 (blue symbols) models during 2003-2008, and (f-j) from satellite observations and 29 AMIP6 models during 2003-2014, over the (a, f) Peruvian, (b, g) Namibian (c, h) Californian, (d, i) Australian and (e, j) Canarian regions.

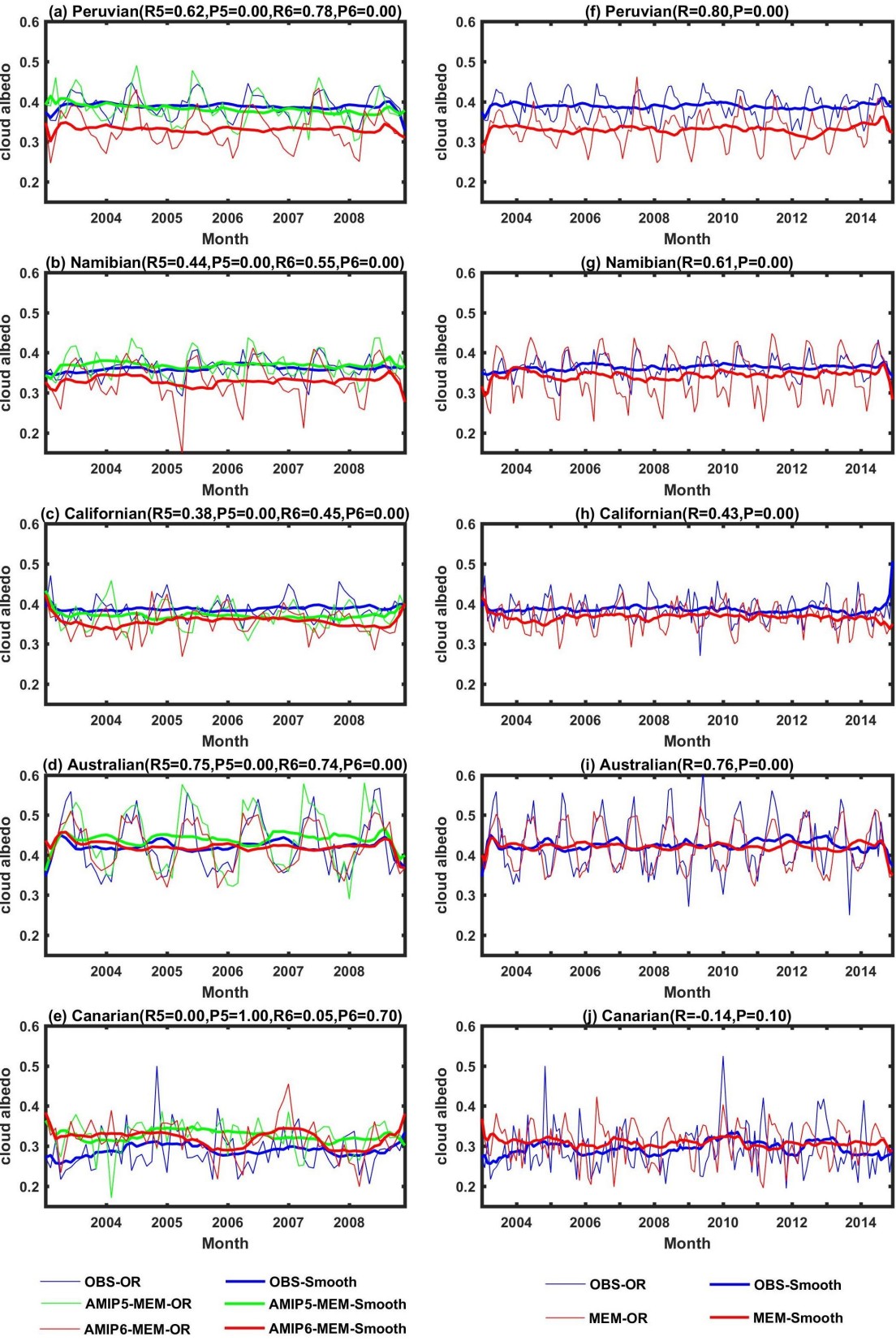

**Figure 3: Monthly mean time series of estimated cloud albedo (a-e) from AMIP5 and AMIP6 multimodel ensemble mean during 2003-2008, and (f-j) from AMIP6 multimodel ensemble mean during 2003-2014 compared with satellite observations, over the (a, f) Peruvian, (b, g) Namibian (c, h) Californian, (d, i) Australian and (e, j) Canarian regions.**

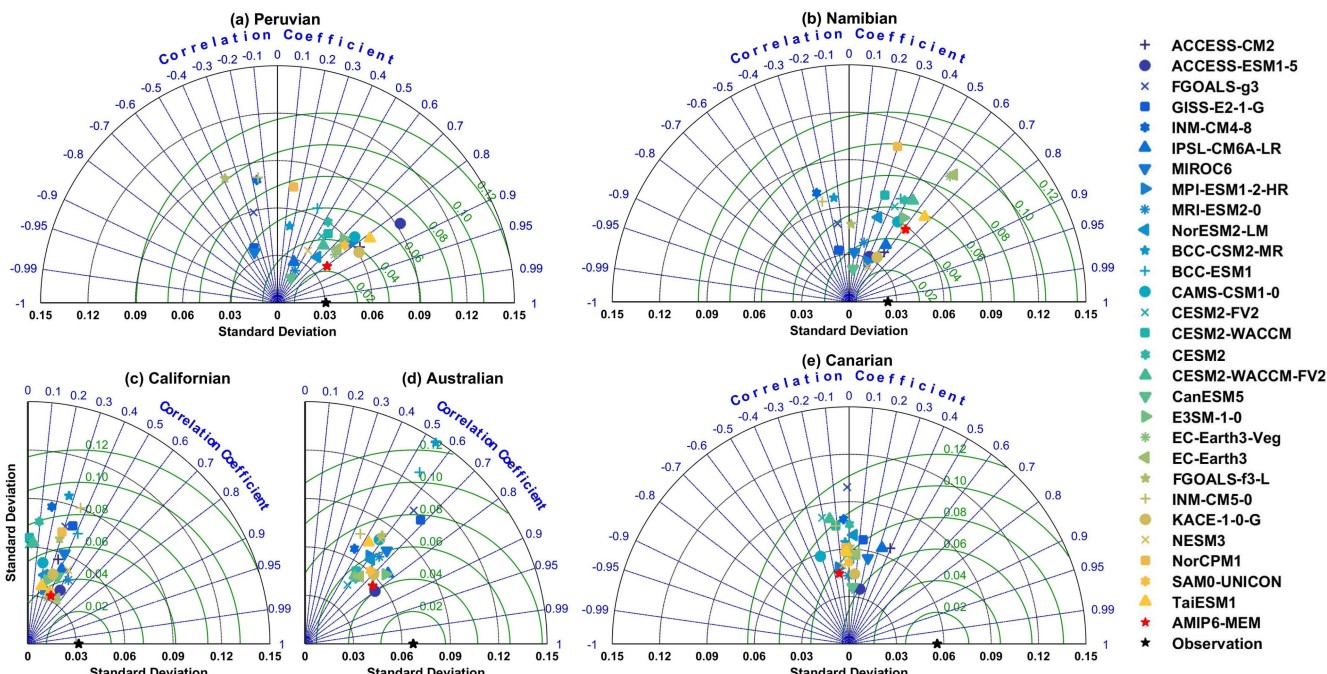

**Figure 4: Taylor diagram for monthly estimated cloud albedo between individual AMIP6 model and satellite observations during 2003-2014 over the (a) Peruvian, (b) Namibian (c) Californian, (d) Australian and (e) Canarian regions. The green circles indicate the centered root mean square error.**

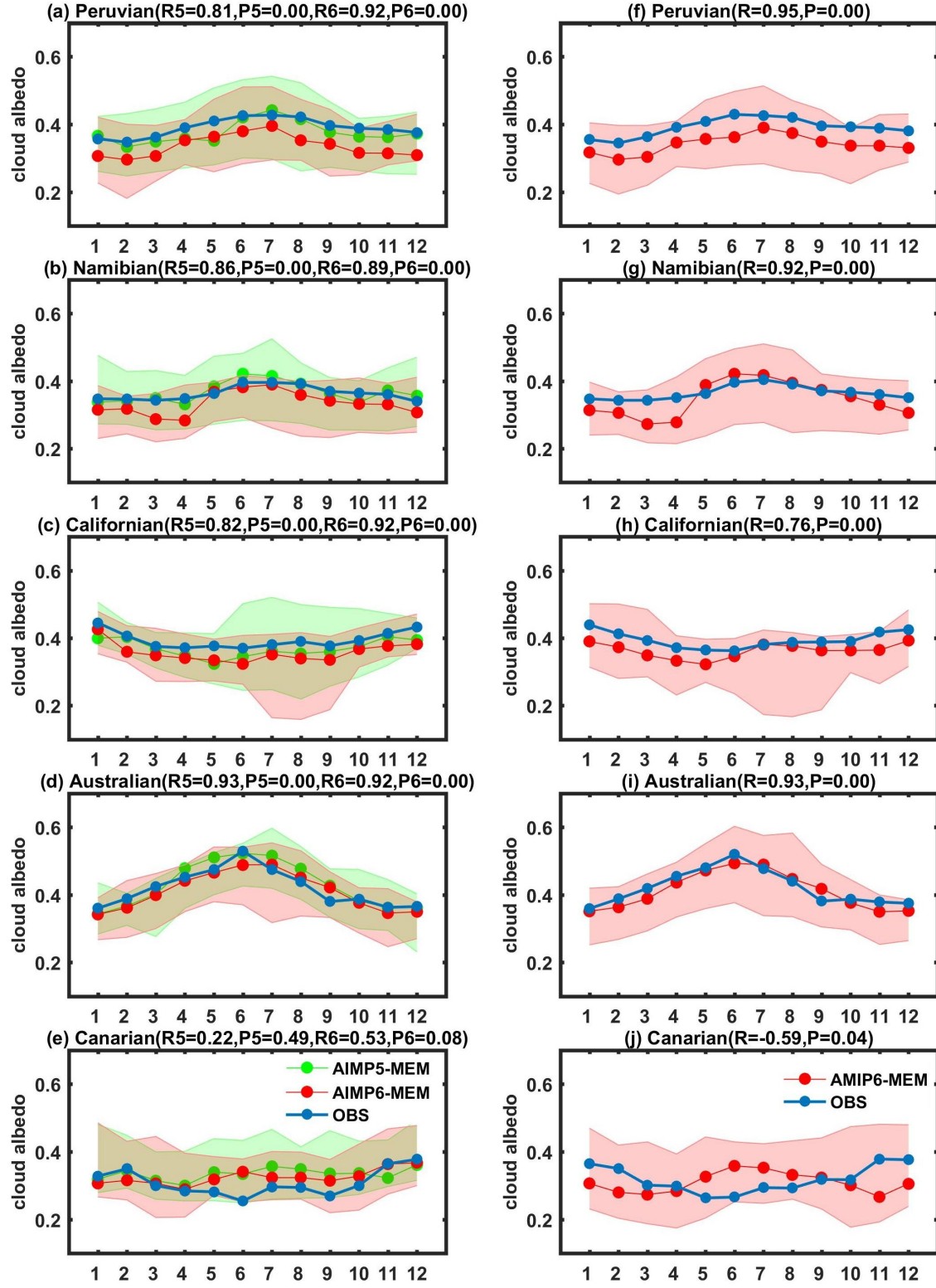

**Figure 5: Annual cycles of the cloud albedo estimated by (a-e) AMIP5 and AMIP6 multimodel ensemble mean during 2003-2008, and (f-j) AMIP6 multimodel ensemble mean during 2003-2014 compared with satellite observations, over the (a, f) Peruvian, (b, g) Namibian (c, h) Californian, (d, i) Australian and (e, j) Canarian regions. The green and red shading areas indicate the range of the cloud albedo simulated by AMIP5 and AMIP6 models, respectively. The temporal correlations (R5/R6/R value) and P5/P6/P value (if P5/P6/P < 0.10, indicating the correlation R5/R6/R is significant) for the seasonal cycles of the cloud albedo obtained from**

**satellite-based observations and models are given in parentheses.**

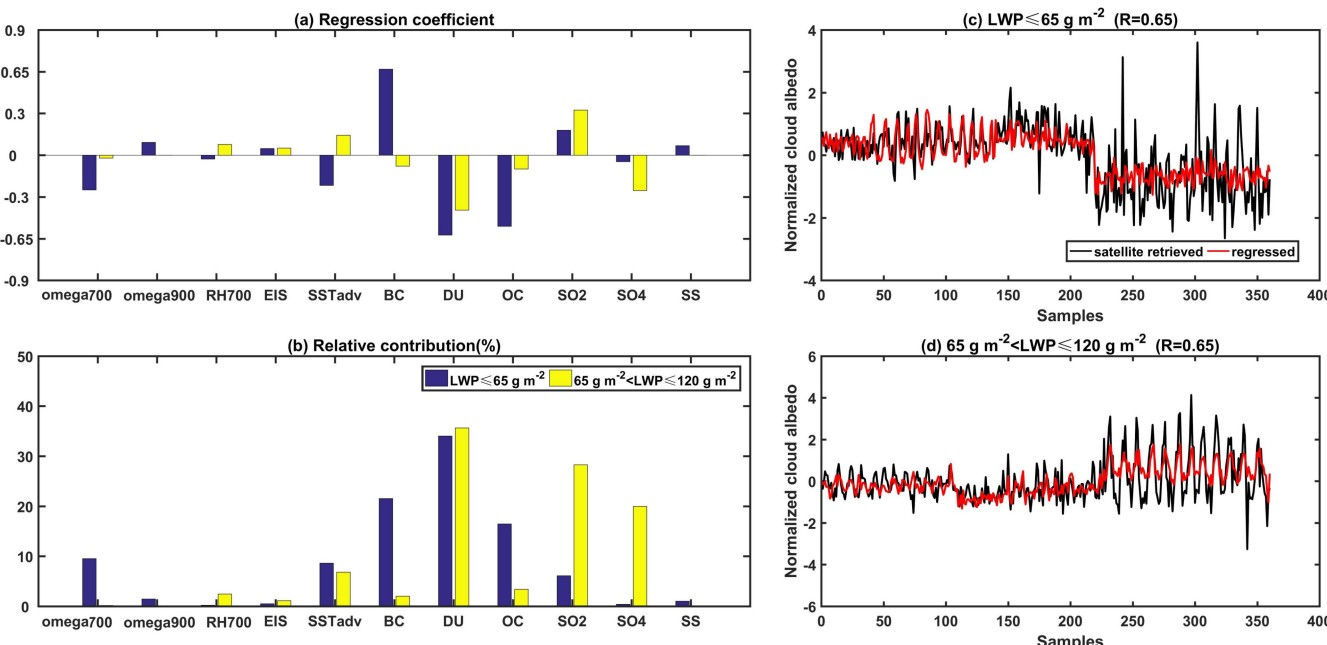

**Figure 6: The (a) regression coefficients and corresponding (b) relative contribution of each predictor variables relating to cloud albedo from the multilinear regression models under two LWP conditions: LWP ≤ 65 g m$^{-2}$ (blue) and 65 g m$^{-2}$ < LWP ≤ 120 g m$^{-2}$**

**(yellow). Note that for ease of comparison, 11 variables are given in the figure, variables without values are not predictive variables of the sample group. And the satellite- and model-driven normalized cloud albedo trained in two sample groups: (c) LWP ≤ 65 g m$^{-2}$ and (d) 65 g m$^{-2}$ < LWP ≤ 120 g m$^{-2}$. The correlations (R value) between satellite- and model-driven normalized cloud albedo are given in parentheses.**