# Peer review of "Evaluation of the CMIP6 marine subtropical stratocumulus cloud albedo and its controlling factors"

_Atmospheric Chemistry and Physics, 2020_

## Referee Comment (RC2)

Review of
**"Evaluation of the CMIP6 marine subtropical stratocumulus cloud albedo and its controlling factors"**
submitted to *Atmospheric Chemistry and Physics* by Jian et al.

This study evaluates the simulation of cloud albedo in subtropical stratocumulus regions in CMIP5 and CMIP6 models and investigates its controlling factors using observations. They find that cloud albedo is underestimated in CMIP6 model AMIP simulations but that, on average, CMIP6 models do a better job than CMIP5 models in reproducing the seasonal cycle of cloud albedo in stratocumulus regions. Lastly, they find differing contributions of various aerosol types to changes in cloud albedo, and that these contributions differ between conditions of low and high liquid water path.

Overall, while certainty there are some interesting results contained in this investigation, there are, in my view, substantial issues with the methodology that need to be addressed. Most importantly, there is an insufficient consideration of the meteorological factors impacting stratocumulus clouds.

Below I provide specific comments.

1. **Insufficient consideration of meteorological factors.** Only two meteorological factors are considered in this study: omega900 and RH850. However, subtropical stratocumulus (including cloud optical depth/thickness, LWP, and cloud droplet effective radius) are impacted by several other important meteorological factors, including sea surface temperature, estimated inversion strength, horizontal surface temperature advection, and wind speed (e.g. Fuchs et al. 2018, Scott et al. 2020, and references therein). Therefore, the consideration of only omega900 and RH850 in this study is inadequate. The omission of the other meteorological factors noted above may in fact greatly affect the results of Section 3.2, "The impacts of different aerosol types and meteorological factors on cloud albedo changes", due to confounding effects. The authors state that "If the correlation between the cloud albedo and a [predictor] candidate is significant at a 90% confidence level, the variable was considered as a predictor factor." But which candidates were considered? Inversion strength and advection have been shown to be the dominant meteorological controls on interannual changes in cloud optical depth in eastern ocean stratocumulus regions (Scott. et al. 2020). Therefore, I find it hard to believe that these two cloud-controlling factors are not significantly correlated with cloud albedo.

   Chen et al. (2014) investigated the effects of aerosols on marine warm clouds using observations. They found that the response of LWP to aerosol loading strongly depends on lower tropospheric stability *and* free-tropospheric moisture. This is additional evidence that the omission of several meteorological factors, especially the inversion strength, is a crucial oversight in the present study.

   Finally, the choices of 900 hPa and 850 hPa as vertical levels for omega and RH are not justified. These levels are not external to the boundary layer. Omega700 (or 500) and

RH700 (or 500) should be used instead, as is standard, since they represent *free-tropospheric* vertical velocity and humidity. The authors should use these vertical levels instead, unless they can provide a compelling justification for their unusual choice of vertical levels.

2. **Lack of analysis of satellite simulator output.** Modern analyses of cloud fraction in GCMs should incorporate at least some analysis of satellite simulator output, such as ISCCP simulator cloud fraction. However, the current study compares the raw GCM cloud fraction with satellite cloud fraction, which is a somewhat outdated approach. Some of the differences between GCMs and the observations found in the paper may be due to different definitions of cloud fraction. The authors do note that "this study… employed the total cloud fractions as there are no available MODIS simulator outputs for CMIP6." However, ISCCP simulator output is available for several CMIP5 and CMIP6 models. MODIS cloud fraction is more comparable to ISCCP simulator cloud fraction than it is to the raw GCM cloud fraction.

3. **Lack of analysis of low-level cloud fraction.** The authors should verify that their key observational results are valid for low-level cloud fraction. The regions chosen are dominated by low clouds, but high cloud variability may impact some of the results.

4. **Lack of verification of results with additional observational data.** MODIS is the state-of-the-art passive satellite cloud dataset, but, given that multi-linear regression can be sensitive to the input data, the authors should examine additional satellite data (such as ISCCP cloud fraction and the Multisensor Advanced Climatology of Liquid Water Path [Elsaesser et al. (2017)]) to corroborate their results and establish robustness. Additional reanalyses should be considered as well for the meteorological data. ERA5 is considered to be the most state-of-the-art reanalysis.

5. How is the threshold of 60 g m$^{-2}$ for LWP chosen?

6. **Excessive detail in discussion of results.** I was quite overwhelmed with the amount of detail discussed in the results section of the paper concerning results for individual models and individual regions and for specific performance metrics. Even after reading the paper a few times, I cannot answer the basic question, "Has the simulation of CMIP6 stratocumulus cloud albedo changed in any major way compared to CMIP5, or is it fundamentally unchanged relative CMIP5?" The paper would be improved by identifying the key differences and similarities between CMIP5 and CMIP6, rather than discussing a detailed and hard-to-remember list of very specific results.

**References:**

Chen, Y. C., Christensen, M. W., Stephens, G. L., & Seinfeld, J. H. (2014). Satellite-based estimate of global aerosol–cloud radiative forcing by marine warm clouds. *Nature Geoscience*, *7*(9), 643-646.

Elsaesser, G. S., O'Dell, C. W., Lebsock, M. D., Bennartz, R., Greenwald, T. J., & Wentz, F. J. (2017). The multisensor advanced climatology of liquid water path (MAC-LWP). *Journal of Climate*, *30*(24), 10193-10210.

Fuchs, J., Cermak, J., & Andersen, H. (2018). Building a cloud in the southeast Atlantic: understanding low-cloud controls based on satellite observations with machine learning. *Atmospheric Chemistry and Physics*, *18*(22), 16537-16552.

Scott, R. C., Myers, T. A., Norris, J. R., Zelinka, M. D., Klein, S. A., Sun, M., & Doelling, D. R. (2020). Observed Sensitivity of Low-Cloud Radiative Effects to Meteorological Perturbations over the Global Oceans. *Journal of Climate*, *33*(18), 7717-7734.

---

## Author Comment (AC1)

**Response to Reviewer #1's Comments:**

Bida Jian et al. (Author)

**We are very grateful for the Reviewer #1' detailed comments and suggestions, which help us improve this paper significantly. Based on the comments and suggestions from the editor and two Reviewers, we add some interpretations and discussions in each section in order to make the manuscript clearer. In particular, some superfluous information in each section is deleted.**

**Please see our point-by-point reply to comments. In addition, all revisions were highlighted in revised manuscript by using track changes.**

**General responses:**

1. Line 11, please change "the subtropical marine subtropical stratocumulus" to "the subtropical marine stratocumulus"

   **Response:** We are sorry to make this mistake. It is corrected in the revised manuscript. Please see the Line:11.

2. Line 12-13, please change "the long-term, monthly and seasonal cycle averaged cloud albedo at five stratocumulus regions were investigated" to "the long-term, monthly and seasonal cycles of averaged cloud albedo at five stratocumulus regions were investigated"

   **Response:** Thanks for your comments. It is corrected in the revised manuscript. Please see the Line: 12-13.

3. Line 16-18, past tense and present tense cannot be used together in a sentence.

   **Response:** It is corrected in the revised manuscript. Please see the Line: 16-18.

4. Line 38, "those of" -> "those over"

   **Response:** It is corrected in the revised manuscript. Please see the Line: 38.

5. Line 50-53, it is only true when considering the cloud albedo for particular clouds that the cloud albedo is determined by COT and solar zenith angle. From a statistical view, it is also strongly

dependent on cloud fraction. In addition, regarding the change of COT with cloud droplet number, size, and liquid water path, a couple references could be mentioned, Zhao et al. (2012, doi:10.1029/2012GL051213) and Xie et al. (2013, doi:10.1175/JCLI-D-12-00517.1). Also, changing "cloud droplets number and sizes" to "cloud droplet number and sizes".

**Response:** We agreed with reviewer. It is corrected and the relevant references are added in the revised manuscript. Please see the line: 53-54.

6. Line 57-59, are you sure these three referred studies are for aerosol-cloud-radiation interaction over subtropical marine stratocumulus regions? In my memory, Twomey (1977) studied the clouds over continent and over the tropical ocean. Anyway, how did you define your study regions here?

**Response:** We are sorry to make this mistake. It is corrected in the revised manuscript. Please see the Line: 57-59. Our study regions are focused on several typical subtropical marine stratocumulus regions (latitude is greater than 10°). Details about the study regions can be found in Section2.

7. Line 67-69, not only the cloud supersaturation, but also other properties (such as aerosol amount, entrainment, detrainment, and so on) would be changed by the dynamical processes. You may read and cite some recent studies to emphasize this point, such as the effect from aerosols and vertical velocity.

**Response:** Thanks for your comments. The sentence is reorganized and relevant references are added in the revised manuscript. Please see the Line: 57-70.

8. Line 90, you may change "addresses" to "provides", "showes" or "gives".

**Response:** It is corrected in the revised manuscript. Please see the Line: 99.

9. Line 94-96, how would you expect the extra errors caused by this kind of interpolation?

**Response:** Thanks for your comments. Indeed, data interpolation will cause extra errors. We calculated the relative bias of monthly cloud albedo between original and interpolation data, the extra errors are about 1% for a single model (see Fig. R1). This indicates that the interpolated data

is valid for this study.

[Figure]

Figure R1: Monthly mean time series of estimated cloud albedo from ACCESS1-0 original (blue line) and interpolation (red line) data during 2003-2008 over Peruvian. The relative bias of monthly cloud albedo between original and interpolation data is given in the title.

10. Line 98, "is required" -> "requires"

    **Response:** It is corrected in the revised manuscript. Please see the Line: 101.

11. Line 101, delete "also" since they are from different product.

    **Response:** Thanks for your comments. It is deleted in the revised manuscript.

12. Line 104-107, a little information about the potential uncertainties from these data process could be helpful.

    **Response:** Thanks for your comments. It is added in the revised manuscript. Please see the Line: 110-113.

13. Line 109-112, two comments I would like to give here. First, you should indicate whether the time is local time or UTC time. Second, regarding the use of the average of two time point cloud observations to represent daily average, it would introduce the time representation error as indicated by Wang and Zhao (2017, doi:10.1002/2016JD025954). This representation error is significant when considering short-term studies (up to 14%), but is neglible when considering long-term statistical analysis. This representation error which is negligible in this study should be

acknowledged.

**Response:** We very thank reviewer for providing detailed comments and suggestions. It is corrected and the relevant references are added in the revised manuscript. Please see the line: 117-120.

14. Line 121-123, As mentioned above, this time representation error could be large for short term, but becomes negligible when considering long-term period.

    **Response:** Thanks for your comments. The comment is added in the revised manuscript. Please see the Line: 132-133.

15. Line 147, delete "for"

    **Response:** It is deleted in the revised manuscript.

16. Line 153-154, why do the authors put this single sentence as a paragraph? Also, I am a little confused about the method described with this sentence, may you please explain a little more?

    **Response:** This sentence is rephrased in the revised manuscript and some extra interpretations are added. Please see below (Line: 175-178):

    "In the study, to avoid influence from seasonal cycle, the long-term mean analyses are implemented with deseasonalized monthly mean data processed by removing a mean seasonal cycle, and then adding the monthly mean value to the interannual anomalies data."

17. Line 158, why 90% instead of 95% confidence level is selected here?

    **Response:** Thanks for your comments. The factors we have selected for this study are supported by physical mechanisms, therefore we consider a confidence level of 90 % is sufficient. In fact, we find that the predictor factor we used here are also significant at a 95% confidence level.

18. Line 160, this is not a complete sentence, you might use "The regression model of cloud albedo $\alpha_{cloud}$ is as follows"

    **Response:** It is corrected in the revised manuscript.

19. Line 198-200, it is still not clear to me how the model NorESM2-LM improve the stratiform cloud parameterization?

    **Response:** Thanks for your comments. Based on the suggestions from Reviewer #2, we delete the superfluous discussions about the individual models in the result section in order to make the results clearer. Thus, the Result (Section 3) are reorganized in the revised manuscript.

20. Line 205, "shows" -> "show"

    **Response:** It is corrected in the revised manuscript.

21. Line 226-230, what do these positive and negative correlations indicate?

    **Response:** In the paper, the positive (negative) correlations indicate that model with a higher mean cloud fraction tend to display a higher (lower) cloud albedo.

22. Line 246-247, please rephrase this sentence.

    **Response:** In the revised manuscript, we reorganize the Result section (Section 3) and the sentence is deleted.

23. Line 260-261, please rephrase the sentence.

    **Response:** In the revised manuscript, we reorganize the Result section (Section 3) and the sentence is deleted.

24. Line 266-267, sorry that I do not understand this sentence, please rephrase it or explain.

    **Response:** It is rephrased in the revised manuscript. Please see the Line: 275-278.

25. Line 272, "the smallest"

    **Response:** It is corrected in the revised manuscript.

26. Line 278-280, how could you explain the large cloud albedos in winter? The following sentences in Line 280-292 did not explain this except they show some influential factors to marine cloud properties.

**Response:** Thank you for your comments. The statistical results show that the cloud albedo is large in winter. We also find that the LWP is generally higher in winters (See Fig. R2). The cloud optical thickness usually increases with an increase in cloud LWP, resulting in an increase in the cloud albedo. However, the underlying physical mechanisms of LWP and cloud optical thickness increase are complex and require more detailed meteorological conditions analysis (e.g., transport of water vapor, relative humidity, high/low pressure system) to explain the result in these regions. Dong et al., (2014) found that at the Azores site (39.098N, 28.038W), the seasonal changes of cloud thickness are link to the seasonal synoptic patterns. They found that the persistent high pressure and dry conditions produce more single-layered clouds during summer, whereas the low pressure and moist air masses during winter generate more total and multilayered clouds, and deep frontal clouds associated with mid-latitude cyclones. During winter the clouds are higher, colder, and thicker with reduced LWP. In order to understand the seasonal variation mechanism over these subtropical marine stratocumulus regions, the analysis of seasonal synoptic patterns is necessary. However, this discussion is beyond the scope of this paper, so we just give the possible influencing factors. We also did some correlation analysis of meteorological factors (e.g., relative humidity at 700hPa, estimated inversion strength and horizontal temperature advection at the surface) and cloud albedo. However, we found that the seasonal change of meteorological factors can't adequately explain the seasonal change of cloud albedo. Thus, we do not consider the results as the content of the manuscript. Please see Figs. R2-R5.

Dong, X., Xi, B., Kennedy, A., Minnis, P., and Wood, R.: A 19-Month Record of Marine Aerosol- Cloud-Radiation Properties Derived from DOE ARM Mobile Facility Deployment at the Azores. Part I: Cloud Fraction and Single-Layered MBL Cloud Properties, J. Clim., 27, 3665-3682, https://doi.org/10.1175/jcli-d-13-00553.1, 2014.

[Figure]

Figure R2: Annual cycles of the estimated cloud albedo and LWP from satellite observations during 2003-2014 over the (a) Peruvian, (b) Namibian, (c) Californian, (d) Australian, (e) Canarian regions. The temporal correlations (R value) and P value (if P<0.10, indicating the correlation R is significant) between cloud albedo and LWP are given in parentheses.

[Figure]

Figure R3: Similar to Fig. R2, but for relative humidity at 700hPa (RH700).

[Figure]

Figure R4: Similar to Fig. R2, but for estimated inversion strength (EIS).

[Figure]

Fig. R5: Similar to Fig. R2, but for horizontal temperature advection at the surface (SSTadv).

27. Line 323-324, what are the potential reasons for the superior performance of AMIP6 at the Australian region?

**Response:** Thanks for your comments. In the revised manuscript, we reorganize the Result sections (Section 3) and the discussion is deleted. The potential reasons for the superior performance of AMIP6 at the Australian region may be due to its better simulation of cloud fraction and planetary albedo. Please see Figs. R6-7. We can see that the AMIP6 performance is

better at the Australian region than other regions, except that Peruvian region.

[Figure]

Figure R6: Seasonal cycles of cloud fraction from AMIP6 MEM outputs (red line) and during 2003-2014 compared with satellite observations (blue line), over the (a) Peruvian, (b) Namibian (c) Californian, (d) Australian and (e) Canarian regions. The R1 indicate the temporal correlations between satellite observations and AMIP6-MEM.

[Figure]

Figure R7: Similar to Figure R5, but for planetary albedo.

28. Line 328-329, these two sentence show almost the same meanings and have been described ealier. You may delete them or one of them to avoid redundant descriptions.

**Response:** It is deleted in the revised manuscript.

29. Line 329-331, regarding the significant role of LWP on the relationship between aerosol and clouds, some recent studies are worthy to read and mention here, Qiu et al. (2017, doi:10.1016/j.atmosenv.2017.06.002) and Zhao et al. (2019, doi:10.3390/atmos10010019).

    **Response:** Thanks for your comments. Related information and reference is already added in the revised manuscript. Please see the Line: 330-333.

30. Line 332-333, you may rephrase this sentence to make it more clear.

    **Response:** It is rephrased in the revised manuscript. Please see the Line: 333-334.

31. Line 341-342, "that the cloud albedo increase with increasing BC and decrease with increasing" -> "that the cloud albedo increases with increasing BC and decreases with increasing"

    **Response:** It is corrected in the revised manuscript. Please see the Line: Line 342-343.

32. Line 355-365, in addition direct effect of dust on cloud properties, the dust aerosol can even further influence the meteorological environment that the clouds form as indicated by Sun et al. (2020, doi:10.1029/2020JD033454)

    **Response:** Related information and reference is already added in the revised manuscript. Please see the Line: 361-364.

33. Line 374-376, do you mean "SO42-"?

    **Response:** It means the "SO4", i.e., the sulfate aerosols.

34. Line 383-384, you cannot compare time scale with region scale. I think what you would like to deliver is the time scale difference between "monthly average" and "instaneous"?

    **Response:** Thanks for your comments. It is corrected in the revised manuscript. Please see the Line: 387.

35. Line 407-408, while this result could be right, personally, I think theoretically the meteorological

factors should have important influence on the interactions between aerosols and cloud albedo for low LWP conditions.

**Response:** We very thank reviewer for providing detailed comments and suggestions. Based on the suggestions from Reviewer #2, we reorganize the Result section (Section 3) and new meteorological factors are considered in the partial correlations analysis in the revised manuscript. We find that the correlations of all aerosol types vary significantly by eliminating the influence of all meteorological parameters in the revised manuscript. The meteorological factors indeed have important influence on the interactions between aerosols and cloud albedo for both low and high LWP conditions.

36. Line 410-414, have you used the data of "SO2"?

    **Response:** The data of "SO2" is used in the study.

---

## Author Comment (AC2)

**Response to Reviewer #2's Comments:**

Bida Jian et al. (Author)

**We are very grateful for the Reviewer #2' detailed comments and suggestions, which help us improve this paper significantly. Based on the comments and suggestions from the editor and two Reviewers, we reorganize the datasets and methods, results and conclusion sections, and add some interpretations in each section in order to make the manuscript clearer. In addition, some superfluous information in each section is deleted.**

**Please see our point-by-point reply to comments. In addition, all revisions were highlighted in revised manuscript by using track changes.**

**General responses:**

1. **Insufficient consideration of meteorological factors.** Only two meteorological factors are considered in this study: omega900 and RH850. However, subtropical stratocumulus (including cloud optical depth/thickness, LWP, and cloud droplet effective radius) are impacted by several other important meteorological factors, including sea surface temperature, estimated inversion strength, horizontal surface temperature advection, and wind speed (e.g. Fuchs et al. 2018, Scott et al. 2020, and references therein). Therefore, the consideration of only omega900 and RH850 in this study is inadequate. The omission of the other meteorological factors noted above may in fact greatly affect the results of Section 3.2, "The impacts of different aerosol types and meteorological factors on cloud albedo changes", due to confounding effects. The authors state that "If the correlation between the cloud albedo and a [predictor] candidate is significant at a 90% confidence level, the variable was considered as a predictor factor." But which candidates were considered? Inversion strength and advection have been shown to be the dominant meteorological controls on interannual changes in cloud optical depth in eastern ocean stratocumulus regions (Scott. et al. 2020). Therefore, I find it hard to believe that these two cloud-controlling factors are not significantly correlated with cloud albedo. Chen et al. (2014) investigated the effects of aerosols on marine warm clouds using observations. They found that the response of LWP to aerosol loading strongly depends on lower tropospheric stability and free-tropospheric moisture. This is additional evidence that

the omission of several meteorological factors, especially the inversion strength, is a crucial oversight in the present study. Finally, the choices of 900 hPa and 850 hPa as vertical levels for omega and RH are not justified. These levels are not external to the boundary layer. Omega700 (or 500) and RH700 (or 500) should be used instead, as is standard, since they represent free tropospheric vertical velocity and humidity. The authors should use these vertical levels instead, unless they can provide a compelling justification for their unusual choice of vertical levels.

**Response:** We are very grateful for the detailed comments and suggestions from reviewer. Indeed, the consideration of meteorological factors in this investigation is insufficient. In the revised manuscript, the relative humidity at 700hPa (RH700),vertical velocity at 900hPa and 700hPa (omega900 and omega700), estimated inversion strength (EIS) and horizontal temperature advection at the surface (SSTadv) are added in the multilinear regression model. Vertical winds below the clouds can affect the exchange between the clouds and the air layer below the clouds, allowing more aerosols to enter the clouds, which serve as CCN, and causing more cloud droplets (Yang et al., 2019). In the study, aerosol mass concentrations at the 910hPa level are employed. Here, the omega900 factor is retained to assess the effect of vertical velocity under the cloud on the cloud albedo. We also add detailed discussions in the revised manuscript. Please see the section 2 and section 3.

Yang, Y., Zhao, C. F., Dong, X. B., Fan, G. C., Zhou, Y. Q., Wang, Y., Zhao, L. J., Lv, F., and Yan, F.: Toward understanding the process-level impacts of aerosols on microphysical properties of shallow cumulus cloud using aircraft observations, Atmos. Res., 221, 27-33, https://doi.org/10.1016/j.atmosres.2019.01.027, 2019.

2. **Lack of analysis of satellite simulator output.** Modern analyses of cloud fraction in GCMs should incorporate at least some analysis of satellite simulator output, such as ISCCP simulator cloud fraction. However, the current study compares the raw GCM cloud fraction with satellite cloud fraction, which is a somewhat outdated approach. Some of the differences between GCMs and the observations found in the paper may be due to different definitions of cloud fraction. The authors do note that "this study… employed the total cloud fractions as

there are no available MODIS simulator outputs for CMIP6." However, ISCCP simulator output is available for several CMIP5 and CMIP6 models. MODIS cloud fraction is more comparable to ISCCP simulator cloud fraction than it is to the raw GCM cloud fraction.

**Response:** We very thank reviewer for providing detailed comments and suggestions. Indeed, the ISCCP simulator output is available for several CMIP5 and CMIP6 models. But the ISCCP output can only provide the monthly mean cloud fraction (variable name: cltisccp) but not corresponding monthly mean shortwave flux. This makes it impossible to calculate the corresponding cloud albedo based on the ISCCP simulator outputs. We tried to compare the AMIP6 cloud fraction with ISCCP simulator cloud fraction, and compared it with satellite observations (See Figure R1). From the correlation coefficients, the results of ISCCP simulator outputs did not outperform the AMIP outputs. There is a good agreement between the ISCCP simulator outputs and AMIP outputs (R3>0.8). In addition, we compared the ISCCP simulator cloud fraction with ISCCP observed cloud fraction (See Figure R2). However, the performance of ISCCP simulator is poor to reproduce the ISCCP observed cloud fraction. Considering that the AMIP outputs have complete corresponding shortwave flux data, we still use the GCM cloud fraction in this study.

[Figure]

Figure R1: Monthly mean time series of cloud fraction from AMIP6 MEM outputs and ISCCP simulator outputs MEM during 2003-2014 compared with satellite observations (MODIS), over the (a) Peruvian, (b) Namibian (c) Californian, (d) Australian and (e) Canarian regions. The R1 and R2 indicate the temporal correlations between satellite observations and AMIP6 outputs and ISCCP simulator outputs, respectively. The R3 indicates the temporal correlations between AMIP6 outputs and ISCCP simulator outputs. Here, the ISCCP MEM is calculated based on 11 model outputs (see the Table R1 below).

[Figure]

Figure R2: Monthly mean time series of cloud fraction from ISCCP simulator outputs MEM (green lines) during 2003-2014 compared with satellite observations (ISCCP, red lines), over the (a) Peruvian, (b) Namibian (c) Californian, (d) Australian and (e) Canarian regions. The R indicate the temporal correlations between satellite observations and ISCCP simulator outputs.

Table R1: The list of CMIP6 models with ISCCP simulator outputs.

|   | Model name | Origin |
|---|---|---|
| 1 | BCC-CSM2-MR | Beijing Climate Center, China |
| 2 | CESM2 | National Center for Atmospheric Research, Climate and Global Dynamics Laboratory, USA |
| 3 | CanESM5 | Canadian Centre for Climate Modelling and Analysis, Environment and Climate Change Canada, Canada |
| 4 | E2SM-1-0 | LLNL, ANL, BNL, LANL, LBNL, ORNL, PNNL and SNL, USA |
| 5 | GFDL-CM4 | National Oceanic and Atmospheric Administration, Geophysical Fluid Dynamics Laboratory, USA |
| 6 | GISS-E2-1-G | NASA/Goddard Institute for Space Studies, USA |
| 7 | IPSL-CM6A-LR | Institut Pierre Simon Laplace, France |
| 8 | MIROC6 | AORI, NIES and JAMSTEC, Japan |

| 9 | MRI-ESM2-0 | Meteorological Research Institute, Japan |
|---|---|---|
| 10 | NorESM2-LM | Norwegian Climate Centre, Norway |
| 11 | TaiESM1 | Research Center for Environmental Changes, Academia Sinica, Taiwan |

3. **Lack of analysis of low-level cloud fraction.** The authors should verify that their key observational results are valid for low-level cloud fraction. The regions chosen are dominated by low clouds, but high cloud variability may impact some of the results.

**Response:** We agree with reviewer. High clouds are a source of uncertainty in this study. However, based on the available data, we cannot exclude its effect and analyze the low cloud albedo independently. As a reference, we used the daytime low-level cloud fraction from CERES SSF1deg product to verify the linear relationship between cloud fraction and planetary albedo (see Figure R3). However, in most regions, low-level cloud fraction didn't reproduce the same linear relationship as total cloud fraction. Charlson et al., (2007) used lidar reflectivity as a proxy for albedo (lidar albedo). They found lidar profiles with high clouds will distort the relationship between integrated attenuated lidar backscatter and low clouds. Fortunately, Bender et al., (2011) quantified the monthly mean regional-scale albedo of marine sratiform clouds based on MODIS and CALIPSO satellite observations, and found that the CALIPSO estimated cloud albedo (excluding the number of profiles with high cloud) is considerable consistent with that of MODIS. Here, the MODIS provided cloud fraction is total cloud fraction. For MODIS, the overlying high clouds didn't contaminate the linear relation. This means that the high clouds have little effect on the calculation of cloud albedo. Considering the consistency of both datasets, we still consider the estimated cloud albedo to be sufficiently representative of the cloud albedo in these stratocumulus regions.

[Figure]

Figure R3: Monthly mean time series of correlation coefficients between total cloud fraction (red lines) and low-level cloud fraction (blue lines) and planetary albedo from CERES observations during 2003-2014, over the (a) Peruvian, (b) Namibian (c) Californian, (d) Australian and (e) Canarian regions. The monthly mean correlation coefficients are given in the title. The left side of the slash ('/') is the monthly mean correlation coefficients of the total cloud fraction, and the left side of the slash ('/') is that of low-level cloud fraction.

Bender, F. A. M., Charlson, R. J., Ekman, A. M. L., and Leahy, L. V.: Quantification of Monthly Mean Regional-Scale Albedo of Marine Stratiform Clouds in Satellite Observations and GCMs, J. Appl. Meteorol. Clim., 50, 2139-2148, https://doi.org/10.1175/jamc-d-11-049.1, 2011.

Charlson, R.J., Ackerman, A.S., Bender, F.A. M., Anderson, T.L. and Liu, Z.: On the climate forcing consequences of the albedo continuum between cloudy and clear air, Tellus B, 59, 715-727, https://doi.org/10.1111/j.1600-0889.2007.00297.x, 2007.

4. **Lack of verification of results with additional observational data.** MODIS is the state-of-the-art passive satellite cloud dataset, but, given that multi-linear regression can be sensitive to the input data, the authors should examine additional satellite data (such as ISCCP cloud fraction and the Multisensor Advanced Climatology of Liquid Water Path [Elsaesser et

al. (2017)]) to corroborate their results and establish robustness. Additional reanalyses should be considered as well for the meteorological data. ERA5 is considered to be the most state-of-the-art reanalysis.

**Response:** Thanks for your suggestions. To verify the sensitive of the results to input data, we employ different datasets to perform the multi-linear regression. Based the ISCCP total cloud fraction (all time) and shortwave flux data (from the ISCCP-H and ISCCP-FH products), we calculated the monthly cloud albedo (see Figures R4-5). However, the ISCCP-H product can't provide daytime total cloud fraction which may introduce errors into the estimated cloud albedo. And the radiative fluxes data provided by ISCCP-FH product is derived from the model output rather than direct observations. In addition, the linear relationship between the cloud fraction and planetary albedo from ISCCP observations is not as stable as that of MODIS and CERES observations (see Figures R4). Considering these uncertainties, the estimated cloud albedo based on ISCCP observations don't consider as inputs in the multiple regression model.

The monthly Multisensor Advanced Climatology of Liquid Water Path (MAC-LWP) is used to test the sensitive of the results to input LWP data. Considering the differences in retrieval methods and values of the MODIS LWP and MACLWP datasets (Greenwald, 2009), we used the threshold of 55 g m$^{-2}$ for MACLWP to better split the samples evenly. The regressed results are given in Figure R6. For comparison, the results from MODIS LWP are also given below (see Figure R7). We can see that the results did not change significantly, indicating that the regressed results are relatively robust. For the reanalyzed dataset, indeed, ERA5 is considered to be the most state-of-the-art reanalysis with higher temporal and spatial resolutions. However, as the aerosol data from MERRA-2, we think it will be better to use corresponding MERRA-2 meteorological data. We also used the ERA5 data to perform the multiple regression model (see Figure R8). Although the results change slightly, the changed results do not affect the main conclusions. Therefore, the MERRA-2 data is used in the revised manuscript.

[Figure]

Figure R4: Monthly mean time series of correlation coefficients between cloud fraction and planetary albedo from CERES/MODIS observations (red lines) and ISCCP observations (blue lines) during 2003-2014, over the (a) Peruvian, (b) Namibian (c) Californian, (d) Australian and (e) Canarian regions. The monthly mean correlation coefficients are given in the title. The left side of the slash ('/') is the monthly mean correlation coefficients of CERES/MODIS observation, and the left side of the slash ('/') is that of ISCCP.

[Figure]

Figure R5: Monthly mean time series of cloud albedo based on ISCCP observation (red lines) during 2003-2014 compared with that of CERES/MODIS observations (green lines), over the

(a) Peruvian, (b) Namibian (c) Californian, (d) Australian and (e) Canarian regions. The R

indicate the temporal correlations between satellite observations and ISCCP simulator outputs.

[Figure]

Figure R6: The (a) regression coefficients and corresponding (b) relative contribution of each

predictor variables relating to cloud albedo from the multilinear regression models under two

MACLWP conditions: LWP ⩽ 55 g m$^{-2}$ (blue) and 55 g m$^{-2}$ < LWP ⩽ 120 g m$^{-2}$ (yellow).

Note that for ease of comparison, eight variables are given in the figure, variables without

values are not predictive variables of the sample group. And the satellite- and model-driven

normalized cloud albedo trained in two sample groups: (c) LWP ⩽ 55 g m$^{-2}$ and (d) 55 g m$^{-2}$

< LWP ⩽ 120 g m$^{-2}$. The correlations (R value) between satellite- and model-driven

normalized cloud albedo are given in parentheses.

[Figure]

Figure R7 (Fig. 6 in the revised manuscript): Similar to Figure R6, but for MODIS LWP

(LWP threshold: 65 g m$^{-2}$).

[Figure]

Figure R8:Similar to Figure R7, but for ERA5.

Greenwald, T. J.: A 2 year comparison of AMSR－E and MODIS cloud liquid water path observations, Geophys. Res. Lett., 36, L20805, https://doi.org/10.1029/2009GL040394, 2009.

5.  How is the threshold of 60 g m$^{-2}$ for LWP chosen?

    **Response:** Thanks for your comments. To maintain a sufficient sample for both groups, the threshold of 60 g m$^{-2}$ for LWP is chose in the previous manuscript. In order to better split the samples evenly, the threshold of LWP was modified to 65 g m$^{-2}$ in the revised manuscript. Hence the sample size for both datasets is 360. In fact, the statistical results from the thresholds of 60 g m$^{-2}$ and 65 g m$^{-2}$ for LWP didn't exhibit obvious differences. Please see Figure R7 (LWP threshold: 65 g m$^{-2}$) and R9 (LWP threshold: 60 g m$^{-2}$).

[Figure]

Figure R9:Similar to Figure R7, but the threshold of 60 g m$^{-2}$ for LWP is used.

6. **Excessive detail in discussion of results.** I was quite overwhelmed with the amount of detail discussed in the results section of the paper concerning results for individual models and individual regions and for specific performance metrics. Even after reading the paper a few times, I cannot answer the basic question, "Has the simulation of CMIP6 stratocumulus cloud albedo changed in any major way compared to CMIP5, or is it fundamentally unchanged relative CMIP5?" The paper would be improved by identifying the key differences and similarities between CMIP5 and CMIP6, rather than discussing a detailed and hard-to-remember list of very specific results.

**Response:** We very thank reviewer for providing detailed comments and suggestions. Based on the comments, we reorganize the Result section (Section 3) and the superfluous information is deleted in the revised manuscript. Please see Section 3.1.